# Analog in-memory computing attention mechanism for fast and energy-efficient large language models

Nathan Leroux [1,4] ✉, Paul-Philipp Manea [2,3,4] ✉, Chirag Sudarshan[2], Jan Finkbeiner [1,3], Sebastian Siegel[2], John Paul Strachan[2,3] & Emre Neftci [1,3]

Transformer networks, driven by self-attention, are central to large language models. In generative transformers, self-attention uses cache memory to store token projections, avoiding recomputation at each time step. However, graphics processing unit (GPU)-stored projections must be loaded into static random-access memory for each new generation step, causing latency and energy bottlenecks. Here we present a custom self-attention in-memory computing architecture based on emerging charge-based memories called gain cells, which can be efficiently written to store new tokens during sequence generation and enable parallel analog dot-product computation required for self-attention. However, the analog gain-cell circuits introduce non-idealities and constraints preventing the direct mapping of pre-trained models. To circumvent this problem, we design an initialization algorithm achieving text-processing performance comparable to GPT-2 without training from scratch. Our architecture reduces attention latency and energy consumption by up to two and four orders of magnitude, respectively, compared with GPUs, marking a substantial step toward ultrafast, low-power generative transformers.

Transformers[1] are central to modern artificial intelligence (AI), powering advances in language models, image processing and beyond. However, their high computational demands lead to substantial energy consumption. Enhancing their efficiency is essential to reduce environmental impact and to keep pace with the exponentially growing size of AI models. The success of transformers as state of the art in sequence processing and generation is enabled by their attention mechanism[2]. To capture dependencies across sequences, the attention mechanism performs dot products between different projections of multiple sequence elements, known as tokens. For generative tasks, the best performance is achieved by autoregressive, decoder-only transformers[3]. At each inference step, the decoder generates a token, which is then appended to the input sequence, forming the input for the subsequent step. To avoid recomputing the keys and values (KV cache) projections of the previously generated tokens, the so-called KV-caching method stores the projections from previous tokens in memory and updates the KV cache with the new projections[4].

In a graphics processing unit (GPU), for each token, the entire KV cache must be transferred from main high-bandwidth memory to cache memory (static random-access memory (SRAM)). In addition, the KV cache is often much larger than the available SRAM memory owing to the dimensions of the stored projections and the sequence length[5]. For instance, the entire KV cache of the model Mistral 7B[6] requires 8 Gb for a batch size of 1, as necessary for inference workloads. In recent technologies, the energy for data access exceeds the energy required for computations[7]. Loading the KV cache for the attention mechanism is thus a major bottleneck, causing increased energy consumption and latency in large language models (LLMs)[8]. To mitigate this bottleneck,

[1]PGI-15, Forschungszentrum Jülich, Jülich, Germany. [2]PGI-14, Forschungszentrum Jülich, Jülich, Germany. [3]Faculty of Electrical Engineering, RWTH Aachen, Aachen, Germany. [4]These authors contributed equally: Nathan Leroux, Paul-Philipp Manea. ✉e-mail: n.leroux@fz-juelich.de; p.manea@fz-juelich.de

a wide body of literature explores resource-efficient algorithms[9]. Alternative architectures to transformers with linear time complexity are developed to improve long-sequence processing efficiency[10,11]. However, transformers continue to exhibit more stable training at scale than alternatives such as Mamba[11], which contributes to their ongoing dominance despite the efficiency of state-space models. Alternatively, different methods have been developed to reduce the memory requirements of KV caching through token pruning[12], latent KV-cache compression[13] or low-rank approximations[14], or by reusing the same KV-cache pairs across multiple heads (grouped-query attention)[15].

While these algorithmic strategies reduce computational and memory overhead, achieving further energy efficiency increasingly depends on hardware innovation. Hardware systems dedicated to specific neural architectures can substantially outperform conventional central processing units and GPUs in terms of energy efficiency[16]. In particular, to mitigate data-transfer overhead of weights loading, several approaches leverage either near-memory or in-memory computing (IMC)[17–21]. IMC is particularly beneficial when using non-volatile memories to store stationary weights in linear layers[22]. However, a full optimization of transformers' inference also requires addressing the attention mechanism, which contributes substantially to the overall computational cost[9,18]. Current IMC solutions do not yet meet all the requirements for efficient hardware implementation of attention. Specifically, KV cache demands fast and energy-efficient memory writing as it is input dependent and must be updated at every generation step. In addition, high parallelism is crucial for low-latency inference, while high memory density is needed for scaling to large models. Finally, long retention time is essential to avoid frequent memory refresh operations. KV cache has been implemented either by dynamic random-access memories (DRAMs)[21,23], which have limited parallelism requiring many digital sequential adders, or by SRAMs[19,24], which are limited by their volatility and relatively low density[25]. Non-volatile memories can be used for linear layers of transformers[17], but are too slow, energy expensive and are not endurant enough for dynamical KV-cache writing[18,22].

In this work, we propose an IMC hardware architecture based on emerging charge-based memory devices, known as gain cells[26,27], to store token projections and compute dot products for the attention mechanism. As a result, gain-cell crossbar arrays simultaneously serve to store the KV cache and to perform attention computation. Gain cells store information in a capacitor, with a dedicated read transistor generating current based on the capacitor's voltage. Unlike DRAM, this enables non-destructive read operations, supporting highly parallel IMC computations. Gain cells have high endurance, fast write speeds and low write energy, and are multi-level. Oxide semiconductor field effect transistor (OSFET)-based gain cells (for example, indium gallium zinc oxide (IGZO) or indium tin oxide (ITO)) are capable of retaining their state for several seconds without a power supply[28–30], can be manufactured with very small feature sizes, achieving higher density than SRAM, and also support three-dimensional (3D) integration, which can further reduce effective area requirements for IMC applications[28–33].

The analog-to-digital conversion required for analog IMC often hinders the advantages this approach offers, as analog-t-digital converters (ADCs) are power and area intensive[34]. To mitigate this issue, charge-based integration is an energy-efficient alternative[35,36]. Here, we choose to perform the core of the attention mechanism—two dot products, scaling and activation function—fully in the analog domains, using charge-to-pulse circuits for activation and inter-module communication, combined with pulse counters for final readout.

Practical applications of LLMs often rely on pre-trained models to reduce training costs. However, our co-optimization approach introduces specific hardware constraints to enhance architectural performance, which leads to a divergence from standard pre-trained models. The multiplications operated with gain cells are non-ideal.

In addition, the normalization in softmax requires summing across all input elements, requiring global connections with an increased hardware complexity scaling with the sequence length[37,38]. In our system, the activation function is instead operated element-wise with charge-to-pulse circuits implementing HardSigmoid functions.

To overcome this discrepancy, we introduce an algorithm that adapts a pre-trained language model to our architecture by scaling each layer according to its statistics and hardware characteristics. With our adaptation algorithm, our model achieves accuracy similar to a pre-trained GPT-2 model without having to train the model from scratch. Overall, the contributions of this study are:

- An in-memory, mixed analog–digital computing design to store token projections and compute attention dot products with gain-cell arrays at high energy efficiency.
- An end-to-end attention mechanism based on analog signals leveraging charge-to-pulse circuits to avoid power- and area-intensive ADCs.
- Quantitative performance analysis of a scalable architecture with area floorplan including analog circuits and digital peripheries.
- A software-to-hardware methodology to map pre-trained (ideal) models to non-traditional hardware reaching an accuracy equivalent to GPT-2.
- Our architecture achieves up to five and two orders of magnitude lower energy consumption and latency, respectively, compared with GPUs.

After detailing the attention mechanism algorithm, we demonstrate its implementation using gain cells and charge-to-pulse circuits. We then show how our approach maps a pre-trained model to our hardware while maintaining high accuracy on common natural language processing (NLP) benchmarks. Finally, we evaluate the architecture's performance in terms of energy consumption, latency and area footprint.

## Results

### Attention mechanism

Figure 1a shows the attention mechanism algorithm. In autoregressive transformers, new token projections called queries ($Q$), keys ($K$) and values ($V$) are created for each inference step from the weights $W_{Q,K,V} \in \mathbb{R}^{D,d}$ and an input token $x_i \in \mathbb{R}^{1,D}$ as:

$$Q_i, K_i, V_i = W_{Q,K,V} x_i, \tag{1}$$

where $i$ is the token index, $D$ is the token dimension and $d$ is the embedding dimension. The keys and values $K_i \in \mathbb{R}^{1,d}$ and $V_i \in \mathbb{R}^{1,d}$ are stored as part of the full KV cache with $K \in \mathbb{R}^{T,d}$ and $V \in \mathbb{R}^{T,d}$, where $T$ is the sequence length. The query $Q_i \in \mathbb{R}^{1,d}$ is not stored but used for inference as

$$S_i = Q_i \cdot K^T; \quad A_i = \phi\left(\frac{S_i}{\sqrt{d}}\right) \cdot V. \tag{2}$$

The dot product between the queries and keys produces an attention score matrix $S_i \in \mathbb{R}^{1,T}$. In standard transformers, the activation function $\phi$ is typically a softmax function, but other nonlinear activation functions can yield similar accuracy[10,39,40]. In particular, sigmoid-based attention has been shown to match softmax-based attention on models up to 7-billion-parameters large[40]. Recent studies show that in the case of sliding window attention[41], the normalization of softmax leads to vanishing memory while sigmoid-based attention can lead to better information[42,43]. The output of the attention mechanism $A_i$ is then obtained by the dot product between the activation $\phi(S_i)$ and the values. In the transformer architecture, multiple attention 'heads' are

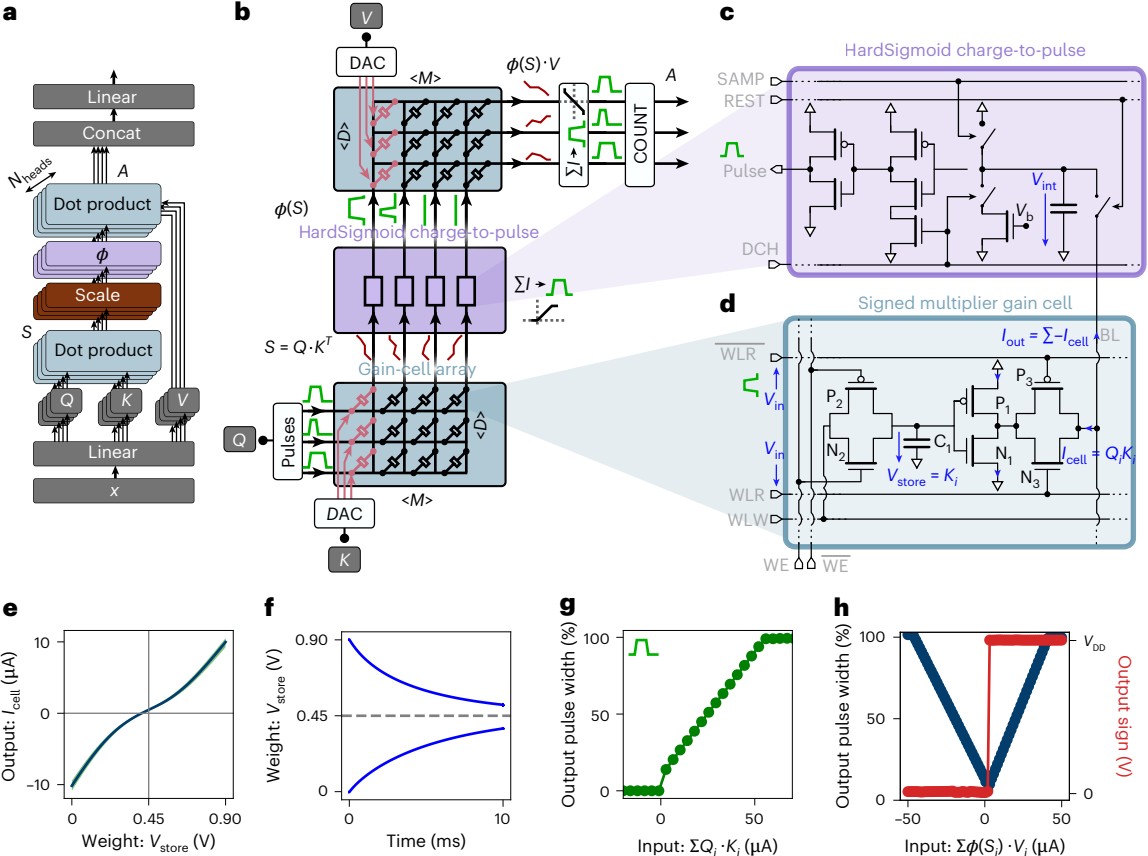

**Fig. 1 | Building blocks of the analog hardware attention mechanism.**
**a**, Multi-head attention architecture. The nonlinear activation is denoted by $\phi$.
Inputs $Q$, $K$ and $V$ are the token projections. $S = Q \cdot K^T$ is the attention score
and $A = \phi(S) \cdot V$ is the final attention output. **b**, Hardware implementation of the
attention mechanism. Red and green traces indicate analog input currents and
generated voltage pulses, respectively. $\sum I$ indicates current integration. $M$ and $D$
denote the sliding window and embedding dimensions. COUNT blocks are pulse
counters returning the digital attention result $A$. **c**, HardSigmoid charge-to-pulse
circuit: integrates bitline (BL) current and emits a pulse width proportional to
the accumulated charge during the discharge phase. The circuit is controlled by
the signals: sample (SAMP), reset (REST) and discharge (DCH) which control
the three states. $V_{int}$, the charge integrated by the charge-to-pulse circuits.

**d**, Signed gain-cell-based multiplier: $V_{store}$ encodes the weight ($K$ or $V$) and is set
via write transistors $N_2$ and $P_2$. $P_1$ and $N_1$ modulate the output current based on
$V_{store}$, while $P_3$ and $N_3$ act as switches driven by the input query $Q$. The signals of the
cell include two complementary word line read (WLR) signals, which serve as the
inputs, a word line write (WLW) signal, a complementary write enable (WE) signal
pair, and a BL that collects the output current. **e**, Simulated output current $I_{cell}$
versus stored voltage $V_{store}$ for $V_{in} = 0.9$ V. Monte Carlo variation bounds are shown
in green. **f**, Simulated voltage decay of the storage capacitor over time due
to leakage from write transistors. **g**, Output pulse width of the HardSigmoid
charge-to-pulse block versus summed input current $\sum_i Q_i \cdot K_i$. **h**, Output pulse
width and sign from the signed charge-to-pulse block versus summed input
current $\sum_i \phi(S_i) \cdot V_i$. All simulations assume $V_{DD} = 0.9$ V.

computed in parallel, concatenated and provided to a subsequent
linear layer to produce the final multi-head attention result.

In decoder-based transformers, causal attention allows the score
matrix $S$ to compare the input token with all previous sequence ele-
ments. However, to prevent the physical memory size from scaling
with the entire sequence length, we employ a type of attention that
is both causal and local: sliding window attention[41]. In this approach,
only a fixed number $M$ of key and value projections are retained in
memory and attention scores for elements older than the last $M$ are
masked (Fig. 2a). Although sliding window attention is local at each
layer, it can still capture global information in deep networks because
the receptive field grows with the number of layers[6].

### End-to-end analog hardware attention
In this section, we first give an overview of how our architecture
performs operations on analog signals to compute attention. Then,
we detail how the different circuits operate. Keys $K$ and values $V$
are stored in two gain-cell arrays. The query $Q_i$ is encoded as pulse-
width modulation (PWM) pulses and is the input of the first array, per-
forming the dot product $Q_i \cdot K^T$. An intermediate charge-to-voltage
pulse block integrates the output currents from the first array and

outputs PWM voltage pulses for the second array, while applying a
HardSigmoid activation function (Fig. 1c). The second array, computing
$\phi(S) \cdot V$ is read out using a signed charge-to-voltage pulse block, where
the resulting pulse widths are measured by a digital counter.

The proposed gain cell, shown in Fig. 1d, contains a write stage
for programming the capacitor $C_1$ and a multiplication stage approxi-
mating the product between the input and the capacitor voltage.

The storage capacitor is charged with a multi-level voltage pulse
emitted by a digital-to-analog converter (DAC). The voltage pulse
is gated to the designated capacitor by a write-enable transmission
gate. Due to leakage in the storage capacitors, the voltages gradually
decay over time. Figure 1f shows the simulated transient response
of the storage capacitor voltage $V_{store}$, which corresponds to the
cell weight for both extreme values 0 V and 0.9 V. An exponential
decay fit of the gain cells leakage reveals that the time constant (that
is, retention time) of our silicon complementary metal–oxide–semi-
conductor (CMOS)-based gain cell is $\tau = 5$ ms. Note that an OSFET-
based gain cell can achieve multiple orders of magnitude longer
retention times[29].

The multiplication stage generates an analog current via a push–
pull transistor pair, with its amplitude set by the stored capacitor

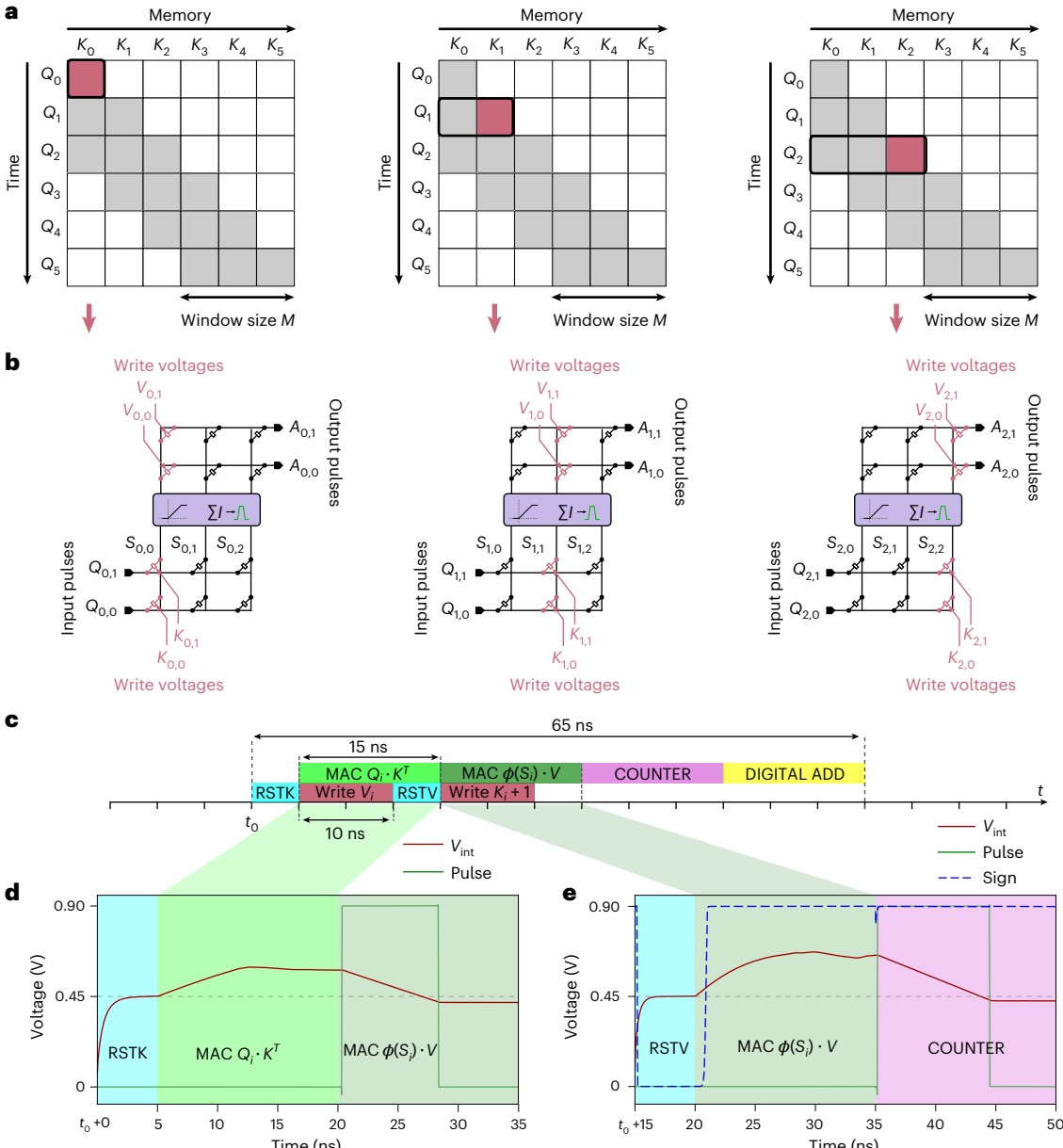

**Fig. 2 | Analog hardware attention pipeline. a**, Three inference steps of a dot product between $Q$ and $K$ in sliding window attention. The gray boxes represent tokens that are attended to and the blank boxes represent the masked tokens. **b**, Equivalent gain-cell-array implementations for an entire attention head. In every inference step, a new column (pointed by red arrows) of $K$ and $V$ is written before the queries $Q$ are applied at the input. $S$ are the currents summed at the bitlines, $\sum I$ represents current integration in charge-to-pulse circuits, and $A$ are the outputs. **c**, Proposed temporal pipeline. The process begins by resetting the charge-to-pulse readout capacitors for the $K$ array (RSTK). While $Q$ pulses

are applied to compute $S = Q \cdot K^T$, the $V$ values for the current token are written in parallel to the $V$ array. After the write, the $V$ readout is reset (RSTV), and the resulting $\phi(S)$ pulses from the charge-to-pulse circuits are applied to the $V$ array to compute $\phi(S) \cdot V$. COUNTER digitizes the final pulse width and sign, and a digital adder combines results from multiple sub-tiles to produce the attention output $A$. **d,e**, Transient simulation of the $\phi(Q \cdot K^T)$ multiply–accumulate (MAC) operation (**d**) and the $\phi(S) \cdot V$ MAC operation including temporal location (**e**). $V_{\text{int}}$ is the charge integrated by the charge-to-pulse circuits, 'Pulse' is their output signal and 'Sign' is the signal current for the counter within the pipeline.

voltage ($V_{\text{store}}$), as shown in Fig. 1e. This current is enabled only during the input pulse, which gates it onto the shared bitline, where currents from multiple cells are summed according to Kirchhoff's law.

In each inference step, both arrays are updated with one column from the key and value matrices, as we will show in more detail in the section 'Analog hardware sliding window attention dataflow'. The $M$ columns of each array represent the $K$ and $V$ of the previous $M$ tokens, while the rows correspond to the $d$ distinct embedding elements.

Due to temporal input encoding, gain-cell outputs also vary over time and must be integrated to compute the dot product.

This is performed by charge-to-pulse circuits (Fig. 1c), which emit PWM voltage pulses. The pulses' width increase linearly with accumulated charge, up to a saturation threshold $S_{\text{sat}}$, as shown in Fig. 1g. The circuit emit pulses only for positive charge, implementing a HardSigmoid activation. Further circuit details are provided in Supplementary Fig. 2.

The pulses representing $\phi(S) \in \mathbb{R}^M$ are fed as inputs to the second gain-cell array to perform the dot product $\phi(S) \cdot V$. A different type of charge-to-pulse circuit integrates the output currents of the second array. Unlike the first one, this signed charge-to-pulse circuit is capable of generating pulses for both positive and negative input

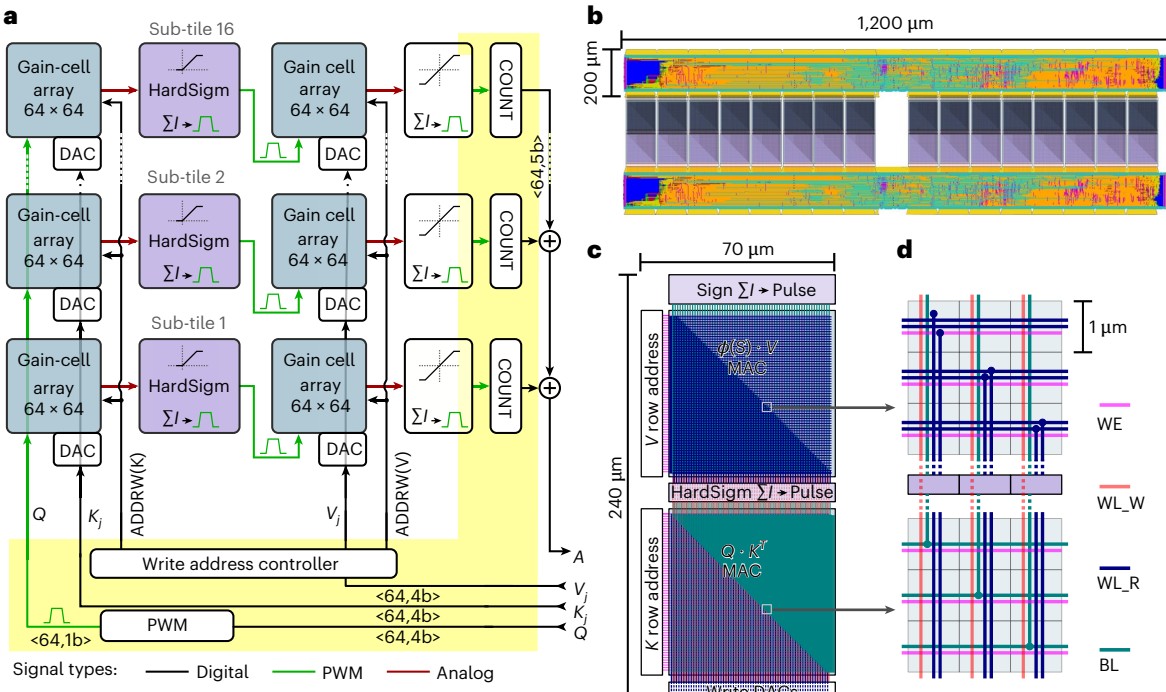

**Fig. 3 | Multi-tile design and layout for multi-head attention. a**, High-level architectural diagram of a hardware unit implementing 1 attention head, supporting a sequence length ($M$) of 1,024 and embedding dimension ($d$) of 64. Labels of the form '<**d**,$p$b>' denote a **d**-element vector with $p$-bit digital precision. The design is partitioned into 16 identical sub-tiles, each integrating two 64 × 64 gain-cell memory arrays to store the $K$ and $V$ projections and perform dot products. Input queries $Q$ are encoded using PWM, while $K$ and $V$ are converted into analog voltages via DACs. A write address controller selects the active memory row for $K$ and $V$ using ADDRW(K) and ADDRW(V). The result passes through a nonlinear activation function (HardSig), and is re-encoded into PWM and routed to the next array within the sub-tile. The final PWM output is digitized using a counter (COUNT) block. Outputs from all sub-tiles are summed by a digital adder to yield the attention result $A$. Digital logic is shown in yellow, PWM signals in green and intermediate analog voltages in red. **b**, Physical layout

corresponding to **a**, showing 16 sub-tiles in the middle with shared digital logic at the top and bottom. Memory arrays are based on compact 6-transistor gain cells, each occupying approximately 1 μm$^2$. The layout is synthesized, placed and routed using Synopsys tools and shown with its default color scheme. **c**, Zoomed-in floorplan of a sub-tile, showing vertically stacked memory arrays, activation blocks and DACs. Blue and green lines indicate input and output signal paths, respectively. **d**, Routing scheme for converting signal orientation between vertical and horizontal. Write DAC signals arrive vertically and connect to vertically oriented word lines (WL_R, blue) in the $Q \cdot K^T$ array. The array's output BLs (green in the bottom array) are routed horizontally. To feed these signals to the vertically stacked HardSigmoid activation block, diagonal wire tapping redirects the horizontal bitlines upward and reorients them for vertical input. The same scheme applies to the $\phi(S) \cdot V$ array. Write enable (WE, pink) and write word lines (WL_W, orange) indicate programmable rows.

charges, while a D flip-flop stores the result's sign. The behavior of this circuit for different inputs is highlighted in Fig. 1h. A 16-level digital counter measures the generated pulse widths and multiplies the result by the retrieved sign bit, resulting in a total precision of 32 levels.

**Analog hardware sliding window attention data-flow**
Having described how inference is performed for one token, we now describe how the architecture processes multiple tokens sequentially. In sliding window attention, the input query is multiplied only with the $M$ most recent keys and values, corresponding to the window size $M$ (Fig. 2a). At each time step, the keys and values must be updated with the most recent token and the oldest one must be forgotten. All other projections remain stationary until they are updated after $M$ cycles. In our implementation, we write the array that encodes the keys and values at inference time in a column-wise manner (Fig. 2b).

Figure 2c illustrates the sequential execution of inference steps in the hardware performing sliding window attention. Read and write operations are interleaved for efficiency, as further detailed in 'Analog sliding window attention timing and execution' in Methods. To perform attention on sliding window sizes and embedding dimensions larger than a single array can support, sub-tiling is used to stack multiple arrays, as shown in Fig. 3, and detailed in 'Sub-tiling to scale attention dimensions' in Methods.

**Pre-trained model hardware-aware mapping and fine-tuning**
Using weights from pre-trained models is challenging because our attention mechanism differs from the conventional ones (Fig. 4a). The main differences are:

- HardSigmoid activation used instead of softmax (Fig. 1b).
- Sliding window attention is implemented instead of causal attention (Fig. 2a).
- Input, stored projections and output are quantized in four, three and five bits, respectively, by digital PWMs, DACs and pulse counters (Fig. 1b).
- Gain-cell arrays are split into sub-tiles before final result summation (Fig. 3a).
- The relation between gain-cell input and stored voltages is nonlinear (Fig. 1e).
- Capacitor leakage causes stored value decay (Fig. 1f).

The implementation of these hardware constraints in our simulations is explained in 'Hardware-based neural network simulations' in Methods. As the nonlinear relation between input voltage and stored voltage in gain cells is described by a third-order polynomial function, this substantially increases the computational complexity and memory requirements to train our gain-cell-based model. Therefore, to adapt the pre-trained public GPT-2 model to our hardware constraints,

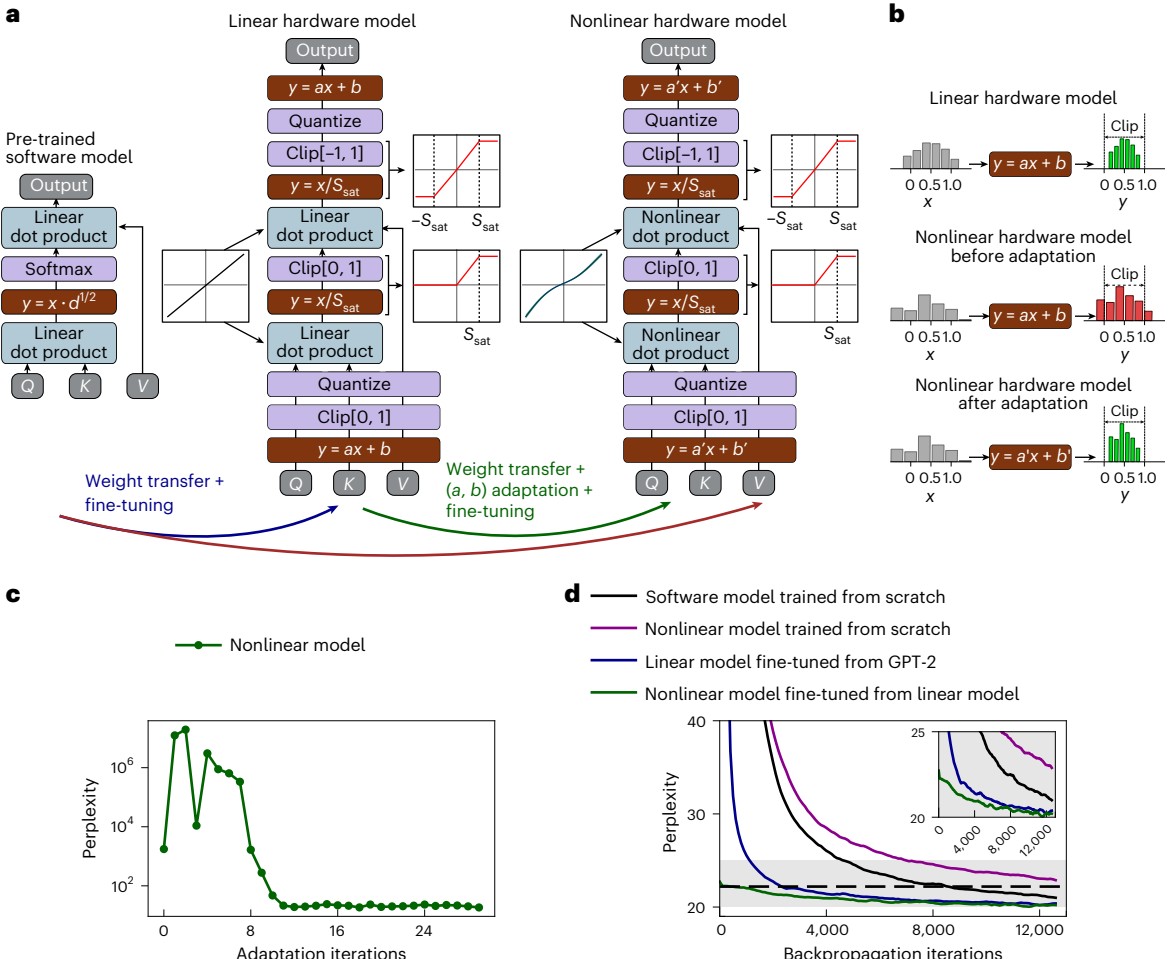

**Fig. 4 | Hardware model adaptation and training. a**, Pre-trained model mapping. $Q$, $K$ and $V$ are the input projections. $d$ is the embedding dimension, $S_{sat}$ is the charge-to-pulse threshold, and $a$ and $b$ are trained scaling parameters. From a software pre-trained model, we fine-tune an intermediate model that integrates all hardware constraints except dot-product nonlinearity. Then, we use a custom adaptation algorithm to map the intermediate model to the gain cell's nonlinearity. Finally, we fine-tune the nonlinear model. **b**, Sketch of the adaptation algorithm for scaling factors. Scaling factors re-scales the input before clipping and quantization. The nonlinear model leads to different statistics (red histogram) than the linear model (green histogram). The adaptation algorithm modifies the scaling factors to match the statistics of the nonlinear model to the statistics of the linear one. **c**, Evolution of perplexity (lower the better) during the adaptation algorithm. **d**, Training curves for the different models. The software model is GPT-2, the nonlinear model is the model with the proposed hardware attention, and the linear model is the hardware attention with ideal linear gain cells. The inset provides a magnified view of the main training curves to emphasize finer differences in model convergence.

we first fine-tune it using an intermediate model. The intermediate model employs ideal linear dot products, but integrates all the other mentioned hardware constraints. The model is trained on predicting the next words of the open-source text collection OpenWebText[44], and the metric used for evaluation is perplexity, which measures the uncertainty of the prediction. In Fig. 4d, we see that our linear intermediate model (blue curve) achieves results equivalent to a public GPT-2 model in less than 3,000 iterations, whereas it takes more than 13,000 iterations for the model trained from scratch (magenta curve). This result shows that performing weight transfer is efficient even though the two models are different (in particular, HardSigmoid activation instead of softmax).

After fine-tuning the intermediate linear model, we transfer the weights to the final hardware model including the gain cell's nonlinearity. This mapping is non-trivial, as all the layers have different statistics, making it difficult to apply a single fit to capture the gain cells' nonlinearity. To circumvent this issue, we introduce scaling operations and an adaptation algorithm described in 'Nonlinear model adaptation algorithm' in Methods. In Fig. 4c, we show how the perplexity of the nonlinear gain-cell model is reduced from 1,757 to 21 during this adaption stage. In Supplementary Fig. 5, we show that this adaptation

algorithm can generalize to other multiplication nonlinearities. After the adaptation algorithm, we can fine-tune the nonlinear model using backpropagation (Fig. 4d, green curve) to further improve the results. The entire process is described in Fig. 4a.

### Downstream task benchmarks
To evaluate the proposed hardware attention mechanism, in Table 1, we benchmark two software baselines and three hardware models on standard language modeling tasks (see details in 'Downstream tasks set-up' in Methods). Our nonlinear hardware model, adapted from a linear baseline and fine-tuned, achieves accuracy comparable to the public GPT-2 model, and equal or better performance than a software model trained from scratch under the same conditions. We further observe that omitting nonlinearity-specific fine-tuning yields near-identical results on most tasks, except LAMBADA and WikiText-2. To test scalability, we apply the same training set-up as GPT-2-XL (1.5 billion parameters). While the hardware version falls slightly short of the public checkpoint, it clearly outperforms the smaller GPT-2 baseline and matches the from-scratch software GPT-2-XL. This indicates that remaining performance gaps are due to

**Table 1 | Downstream task results**

| | ARC-E | ARC-C | WinoGrande | HellaSwag | LAMBADA | LAMBADA | PIQA | WikiText-2 | Average | Average |
|---|---|---|---|---|---|---|---|---|---|---|
| | acc ↑ | acc ↑ | acc ↑ | acc ↑ | ppl ↓ | acc ↑ | acc ↑ | ppl ↓ | acc ↑ | ppl ↓ |
| Public software model | 43.81 | 22.70 | 51.62 | 31.14 | 35.15 | 45.96 | 62.89 | 37.37 | 43.02 | 36.26 |
| Software model trained from scratch | 42.34 | 23.46 | 50.20 | 29.73 | 46.39 | 41.56 | 61.48 | 41.25 | 41.46 | 43.82 |
| Linear hardware model | 42.80 | 23.46 | 52.41 | 30.31 | 51.69 | 38.10 | 61.21 | 39.79 | 41.38 | 45.74 |
| **Nonlinear hardware model with adaptation** | **42.09** | **22.87** | **50.51** | **30.10** | **76.59** | **31.61** | **61.53** | **42.34** | **39.79** | **59.47** |
| **Nonlinear hardware model with adaptation and fine-tuning** | **43.94** | **22.78** | **51.14** | **30.18** | **43.08** | **40.16** | **62.62** | **39.97** | **41.80** | **41.52** |
| Public software model-XL | 58.29 (+14.48) | 28.50 (+5.80) | 58.33 (+6.71) | 50.89 (+19.75) | 9.68 (−25.47) | 63.87 (+17.91) | 70.84 (+7.95) | 20.38 (−16.99) | 55.12 (+12.10) | 15.03 (−21.23) |
| Software model trained from scratch-XL | 53.82 (+11.48) | 25.76 (+2.30) | 53.75 (+3.55) | 42.54 (+12.81) | 14.82 (−31.57) | 56.33 (+14.77) | 68.71 (+7.23) | 24.98 (−16.27) | 50.15 (+8.69) | 19.90 (−23.92) |
| Linear hardware model-XL | 54.08 (+11.28) | 27.47 (+4.01) | 57.93 (+5.52) | 45.51 (+15.20) | 12.32 (−39.37) | 58.54 (+20.44) | 68.01 (+6.80) | 23.26 (−16.53) | 51.92 (+10.54) | 17.79 (−27.95) |
| **Nonlinear hardware model-XL** | **53.79 (+9.85)** | **27.30 (+4.52)** | **54.70 (+3.56)** | **46.70 (+16.52)** | **12.17 (−30.91)** | **59.48 (+19.32)** | **68.17 (+5.55)** | **22.29 (−17.68)** | **51.69 (+9.89)** | **17.23 (−24.29)** |

The metrics are the percentage of accurate word predictions (acc), and the perplexity (ppl), a measure of prediction uncertainty. The last two columns average the accuracy results and the perplexity results for each model, respectively. Values in parentheses (±x) indicate the improvement of XL models relative to their smaller counterparts (GPT-2-XL results – GPT-2 results). Rows in bold correspond to our results.

training iteration differences (the number of iterations for the public model is undisclosed), not hardware limitations.

## Circuit computing accuracy

The accuracy of our circuits for attention computation is highlighted in Fig. 5a,b. For each of the two dot products, we simulate one 64 × 64 array and the corresponding 64 charge-to-pulse circuits. The results of the first dot product, which are shown in Fig. 5a, are fed as input to the second dot product and are shown in Fig. 5b. For each plot, we compare the simulations performed with SPICE (a circuit simulation software) with the model used for neural network simulations.

## Energy consumption and latency

The circuit's operational speed and timing, on which the energy assumptions are based, are shown in Fig. 2d. The total latency of attention can be estimated to 65 ns.

The gain-cell arrays and charge-to-pulse circuits consume 1,120 pJ per token computation for the first dot product, and 700 pJ for the second dot product. The lower energy consumption in the second dot-product arrays is attributed to the sparser activation of its input $\phi(S)$, leading to less current in the second gain-cell array. The digital control and routing block consumes a total power of 113.7 mW, or 4 nJ per token, while the DACs require 330 pJ. Overall, we can estimate the power consumption of processing 1 token for 1 attention head to 6.1 nJ. A pie chart of the power composition attributed to each unit is shown in Fig. 5e.

The energy and latency of our architecture, compared with three different GPUs, are shown in Fig. 5c,d. Focusing on the attention mechanism alone, our architecture can lead to a speed-up of ×7,000 compared with Nvidia Jetson Nano, ×300 compared with Nvidia RTX 4090 and ×100 compared with Nvidia H100, as well as an energy reduction of ×40,000 compared with Jetson Nano, ×90,000 compared with RTX 4090 and ×70,000 compared with H100.

## Area and floorplan

On the basis of our assumptions, described in 'Area estimation' in Methods, for the worst-case scenario, the area of the proposed gain cell

is 1 μm$^2$. Figure 3c shows the floorplan of a single tile, which includes 64 shared DACs for writing the weights, 2-row address decoders and charge-to-pulse circuitry. The total area of 1 head, shown in the floorplan in Fig. 3b, is 500 × 10$^{-3}$ mm$^2$ including digital control circuitry.

However, other studies have demonstrated substantially smaller gain-cell dimensions[45]. On the basis of this, and following the methodology outlined in 'Area estimation' in Methods, we estimate that the area of the gain-cell crossbars required for the entire GPT-2 attention-head KV cache is approximately 15.7 × 10$^{-3}$ mm$^2$, excluding digital control circuitry.

In Supplementary Fig. 7, we show that multiple attention heads can be executed using parallel tiles on-chip and stacked in 3D with multiple layers, sharing peripheral and digital logic. As discussed in 'Area estimation' in Methods, 3D stacking can further improve area efficiency. On the basis of ref. 45, we estimate the total area required for a GPT attention-head KV cache, excluding digital control, to be $\frac{36.7}{N} \times 10^{-3}$ mm$^2$, where $N$ denotes the number of vertical stacks. The resulting area is:

- 36.7 × 10$^{-3}$ mm$^2$ for $N = 1$
- 9.2 × 10$^{-3}$ mm$^2$ for $N = 4$
- 4.6 × 10$^{-3}$ mm$^2$ for $N = 8$
- 3.1 × 10$^{-3}$ mm$^2$ for $N = 12$

## Discussion

In this work, we proposed an analog IMC architecture addressing the energy consumption and latency bottlenecks of the attention computations at the core of generative AI models.

Our design leverages capacitor-based gain cells, offering an efficient solution for both memory storage and computation, substantially improving energy efficiency and speed. To avoid power-intensive ADCs, we perform the attention computation in the analog domain, using charge-to-pulse circuits to transmit analog signals between computation stages. This approach introduces non-ideal operations compared with digital attention computations, but with substantial efficiency gains. Another contribution is a hardware-aware

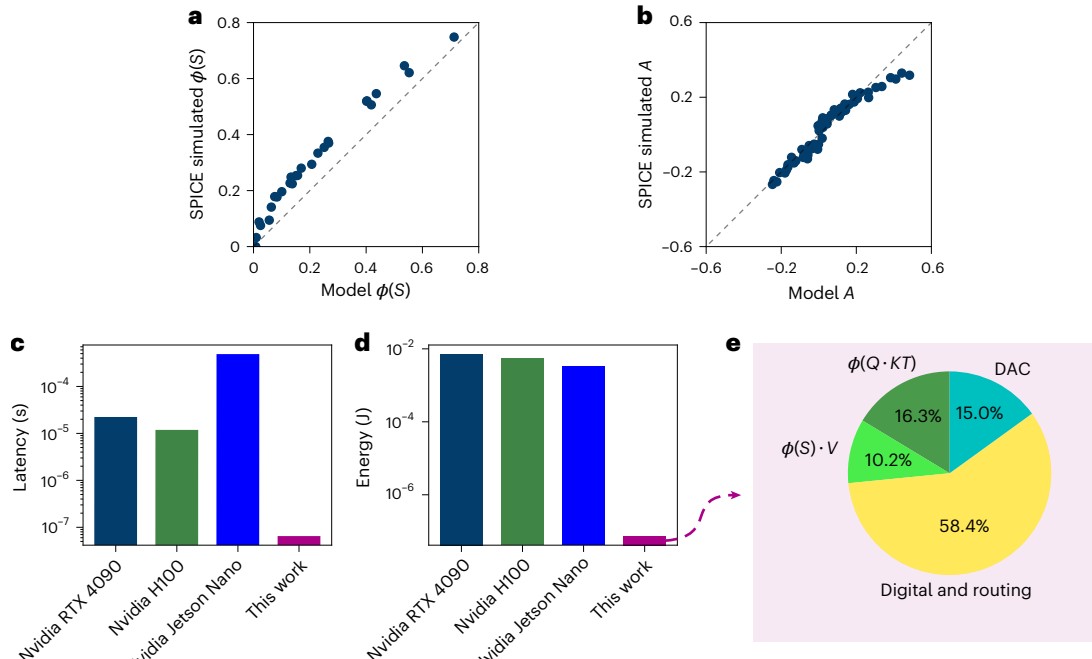

**Fig. 5 | Analog hardware attention mechanism accuracy and performances.** **a**, Comparison of expected results model versus SPICE simulation results for the charge-to-pulse circuit output $\phi(S)$ with $S = Q \cdot K^T$ the results of the first crossbar array and $\phi$ the transfer function of the charge-to-pulse circuit. **b**, Comparison of PyTorch model versus SPICE simulation results for the second crossbar array output $A = \phi(S) \cdot V$. **c,d**, Latency and energy consumption per token of the attention mechanism for 1 processed token (**c**) and energy consumption for a 12-head attention mechanism implemented by a consumer GPU, a server GPU, an embedded application-specific GPU and our hardware architecture (**d**). **e**, Energy consumption ratio for the different modules of our hardware architecture, including analog and digital signals.

training methodology compensating for the circuit non-idealities. Nonetheless, future circuit optimizations could further reduce any discrepancies.

Our neural network simulations confirm that an LLM implemented with our hardware attention achieves results comparable to software-based networks, even on complex NLP tasks. Nonetheless, our larger network slightly underperforms the baseline, and therefore deeper neural network training will require further methods to mitigate the vanishing gradient issue due to clamping values. This slight performance gap should still be put in perspective with the reduced energy consumption. While our study uses device-level simulations to evaluate design performance, our adaptation algorithm demonstrates potential for measured device implementations, as it allows most of the training process to proceed without requiring precise device-specific models of nonlinear behavior, making the approach generically applicable and computationally efficient.

Our architecture can benefit from OSFET transistors that enable dense 3D integration[45,46]. Moreover, the KV-cache size grows modestly compared with the overall models' parameters count[14,15,47]. Our system could therefore be applied to larger networks with a moderate area footprint. Latency is reduced by up to two orders of magnitude, and energy consumption by up to four orders for attention computations alone compared with GPUs. While we focus on the attention mechanism, a major bottleneck in generative transformers' inference, substantial reductions in overall energy consumption require optimizing all components. In the future, our hardware attention mechanism can be integrated with other IMC techniques to implement low-power linear layers.

In conclusion, this work demonstrates hardware-algorithm co-optimization achieving low latency and energy consumption while maintaining high model accuracy. In addition, it highlights the promise of IMC with volatile, low-power memory for attention-based neural networks, marking an important step toward ultrafast, energy-efficient generative AI.

## Methods

### Hardware-based neural network simulations

We implement the sliding window attention by masking the elements of $S$ outside the sliding window (blank spaces in the example Fig. 1). The HardSigmoid charge-to-pulse circuit is modeled by the equation

$$\phi(S) = \begin{cases} T_{max} & \text{if } S \geq S_{sat} \\ \frac{T_{max}}{S_{sat}} S & \text{if } 0 < S < S_{sat} \\ 0 & \text{if } S \leq 0 \end{cases} \quad , \tag{3}$$

where $T_{max} = 15$ ns is the maximum pulse length for the input pulse generators. The input queries $Q$ are quantized in 16 levels between 0 and 1, the stored $K$ and $V$ projections are quantized in 8 levels between 0 and 0.9, and the outputs of the second dot product are quantized in 32 levels between −1 and 1. The quantized models (linear intermediate hardware model and nonlinear hardware model) are trained with quantization aware training[48]: quantization is done only in the forward pass and the backward pass is done in full precision.

For the nonlinear model of the gain cell, the third-order polynomials

$$S = \sum_i^3 \sum_j^{3-i} Q \cdot \left( K^T - K_{offset} \right)^i V_{in}^j C_{i,j}$$

$$A = \sum_i^3 \sum_j^{3-i} \phi(S) \cdot \left( V - V_{offset} \right)^i V_{in}^j C_{i,j} \tag{4}$$

are used with $S$ and $A$ as the outputs, $Q$ and $\phi(S)$ the input pulse width, $K$ and $V$ the stored voltage, the constant $V_{in} = 0.9$ V is the input voltage of the cell applied at the word line read (WLR) ports, the constant $y_{offset} = 0.45$ V corresponds to half the supply voltage ($V_{DD}/2$), and $C_{i,j}$ as fit parameters from the curve Fig. 1e. To speed-up computation

during training, we compute all the tokens in parallel with $Q \in \mathbb{R}^{T,D}$, $K^T \in \mathbb{R}^{D,T}$, $V \in \mathbb{R}^{T,D}$ and $\phi(S) \in \mathbb{R}^{T,T}$ (the batch dimension and the head dimension are omitted for simplicity).

The capacitor leakage leads to an exponential decay in the stored value. After discretization, the exponential decay is formulated as

$$y_t = y_{t-1} e^{-\frac{\Delta_t}{\tau}}; \quad \Delta_t = L\delta_t, \tag{5}$$

where $\tau$ is the time constant of the capacitors, $\Delta_t$ is the time elapses between two inference steps, $\delta_t$ is the latency caused by each neural network layer, and $L$ is the number of layers. To model the decay of all capacitors at all time steps in parallel, we introduce a decay mask $\alpha \in \mathbb{R}^{T,T}$ defined as

$$\alpha = e^{-\frac{\Delta_t}{\tau} m_{t,t'}}; \quad m_{t,t'} = \max(0, t - t'), \tag{6}$$

where $m$ is the relative tokens' position. To optimize computation, the decay mask is directly integrated in the dot-product computation as

$$S = \sum_i^3 \sum_j^{3-i} \left( Q \cdot (K^T - K_{\text{offset}})^i V_{\text{in}}^j C_{i,j} \right) \alpha^i$$
$$A = \sum_i^3 \sum_j^{3-i} (\phi(S)\alpha^i) \cdot (V - V_{\text{offset}})^i V_{\text{in}}^j C_{i,j} \tag{7}$$

In our simulation, we chose a time constant $\tau = 5$ ms to be consistent with the data from Fig. 1h. We chose $\delta_t = 65$ ns to be equal to the latency of our full hardware attention mechanism (Fig. 2c). Our decay factor is therefore $\frac{\Delta_t}{\tau} = \frac{12 \times 65 \times 10^{-9}}{5 \times 10^{-3}} \simeq 1.6 \times 10^{-4}$. In a full transformer implementation, the latency per layer $\delta_t =$ will be higher than 65 ns as it will also include latency from other modules, such as feedforward neural networks. However, time constant $\tau$ of three orders of magnitude larger were reported in OSFET-based gain-cell memories[26,29], and therefore we conclude that the choice of decay factor of $1.6 \times 10^{-4}$ is very conservative. In Supplementary Fig. 6, we study empirically the effect of the decay constant over language processing accuracy. It is noteworthy that the decay of stored keys and values may not necessarily hinder network performance: several approaches in deep learning leverage exponential decay masks to enhance memory structure[39,49]. In Supplementary Information section 'Effect of capacitor's leakage', we study the connection between the KV pairs decay and the relative positional embedding called AliBi[49].

To speed up our training process, we used the library Triton[50] to incorporate our simulations into an adapted version of the flash attention algorithm[51], which optimizes the GPU resources. This method led to a factor of five latency reduction during training.

For the adaptation, the algorithm was repeated until the mean and standard deviation of the output of the scaling functions of the nonlinear model matches the mean and standard deviation of the linear model within a tolerance ratio: $|\sigma_{\text{NL}} - \sigma_{\text{L}}| < 0.0001$ and $|\mu_{\text{NL}} - \mu_{\text{L}}| < 0.0001$.

### Nonlinear model adaptation algorithm

$$y = ax + b \tag{8}$$

with distinct scalars $a$ and $b$ for each of the $Q$, $K$ and $V$ projections, as well as for the output of the attention, with separate factors applied across different attention heads and layers.

To choose the scaling parameters $a$ and $b$, we develop an algorithm inspired by ref. 52, detailed in Supplementary Algorithm 1. Given a set of input samples, we use an iterative loop that updates the scaling parameters so that the output of the scaling function of the nonlinear model matches the statistics of the linear model (as sketched in Fig. 4b). First, we measure the standard deviation $\sigma_{\text{L}}$ and the mean $\mu_{\text{L}}$ of the output

of every scaling stage (see equation (8)) of the linear model on a large set of samples. Then, at each iteration, we measure the standard deviation $\sigma_{\text{NL}}$ and the mean $\mu_{\text{NL}}$ for the scaling stage of the nonlinear model. For each iteration, the scaling parameters are updated as

$$\begin{aligned} a &\leftarrow a \frac{\sigma_{\text{L}}}{\sigma_{\text{NL}}} \\ b &\leftarrow b + (\mu_{\text{L}} - \mu_{\text{NL}}) \end{aligned}. \tag{9}$$

### Analog sliding window attention timing and execution

To support efficient sequential inference, our architecture implements sliding window attention using a pipelined read–write mechanism across analog gain-cell arrays. At each inference step, new $(K, V)$ pairs are written into the arrays while the current query $(Q)$ is applied, ensuring that memory access and computation overlap.

Each attention step begins with a 5 ns discharge phase to reset the storage capacitors of the gain cells. New $K$ and $V$ vectors are written to a column of the respective arrays using 10 ns multi-level voltage pulses generated by 3-bit DACs. In parallel, the input query $Q$ is encoded as PWM voltage pulses with durations between 0 ns and $T_{\max} = 15$ ns, generated by 4-bit (16 levels) voltage pulse generators operating at 1 GHz.

This parallelization is possible because the $V$ array is not required during the $Q \cdot K^T$ computation phase and can therefore be updated while the first dot product is processed. Once the write is complete, the charge-to-pulse circuit for the $V$ array is reset, and the resulting $\phi(S)$ pulses from the $K$ array's readout are applied to the $V$ array to compute the second dot product $\phi(S) \cdot V$.

After $M$ time steps, when all columns in the $K$ and $V$ arrays have been populated, the first column is overwritten, preserving a sliding attention window of fixed size $M$. The succession of write and read phases implements a sequential sliding window attention mechanism, with minimal idle time and continuous throughput. This pipelined execution scheme is visualized in Fig. 2c, and forms the basis for the latency and energy analysis presented in later sections.

### Sub-tiling to scale attention dimensions

IR drop, caused by resistive losses in interconnects, results in reduced accuracy in large-scale analog crossbar arrays[53]. To mitigate IR drop issues, we limit the size of our gain-cell arrays to 64 × 64. However, most NLP applications require larger either a larger window dimension $M$ (columns) or a larger embedding dimension $d$ (rows). To accommodate larger dimensions, we perform inference across multiple sub-tiles, as shown in Fig. 3a.

In this paper, we implement a GPT-2 model with an embedding dimension $d = 64$ and a sliding window size $M = 1,024$. Therefore, the entire KV cache of the window size $M$ is divided into 16 sub-tiles, each having its charge-to-pulse blocks and storing a fraction of the $K$ and $V$ in two 64 × 64 arrays. A write address controller keeps track of the current write index. All tiles receive the same input $Q$ generated by the digital block in parallel, are measured by pulse counters and summed by 64 digital adders, each with 16 inputs (Fig. 3b,c). In sliding window attention, the maximum attention span is equal to $L(M-1) + 1$ (ref. 43). Therefore, in the presented architecture, the maximum attention span can be increased by increasing the number of sub-tiles. However, this leads to additional area footprint scaling linearly with the sliding window dimension, and additional latency as each digital adder requires one clock cycle.

### Hardware-based neural network training

To evaluate our training algorithm and the inference accuracy of our architecture, we implement the analog gain-cell-based attention mechanism on the GPT-2 architecture[54]. GPT-2 is a transformer neural network with 124 million parameters, 12 layers, an attention mechanism input dimension of 768, 12 heads per attention block

and a head dimension of 64. We used the open-source text collection OpenWebText[44] split between training and testing samples, and the pre-trained GPT-2 tokenizer to encode the plain text into tokens (vectors of size 50,304 each). Each training iteration had a batch size of 1,920, with sequences of length 1,024 per sample. We selected a sliding window size of 1,024, which matches the number of gain-cell rows in the memory. As the sequence length also equals 1,024, each gain cell is written only once per sequence, eliminating the need to overwrite cells during one sliding window iteration. For a larger sequence length, the gain cells would be overwritten, as described in the section 'Analog hardware sliding window attention data-flow'. To train the network, the next token in the sequence is predicted for each input token. Thus, the target sequences are the input sequences shifted by one token. The cost function used was cross-entropy, calculated between the predicted sequence and the target sequence. We used backpropagation with the AdamW optimizer[55], with a learning rate of $6 \times 10^{-4}$ and a weight decay of 0.1. The results of each evaluation are averaged over 4,000 samples.

### Downstream tasks set-up

The datasets cover various types of problem. Our benchmarking set-up is inspired by refs. 11,56 in terms of evaluated tasks and metrics. ARC-Easy and ARC-Challenge[57] focus on question answering, with ARC-Easy containing straightforward questions and ARC-Challenge featuring more difficult ones. WinoGrande[58] evaluates common-sense reasoning and co-reference resolution by presenting minimal pairs to resolve ambiguities. HellaSwag[59] tests common-sense inference, requiring models to predict the most plausible continuation of a given context. LAMBADA[60] evaluates models' text understanding through a word prediction task that requires comprehension of broader discourse, not just local context. PIQA[61] assesses physical common-sense reasoning, testing a model's understanding of physical scenarios. WikiText-2[62] is a general text corpus derived from Wikipedia articles to assess long-term dependencies processing, text prediction and generation capabilities. For WikiText-2, we report perplexity scores normalized by the word count in the original text. For fair comparisons, except for software public GPT-2, all the models were evaluated after the same number of training iterations. The linear hardware model was trained on 13,000 iterations, the nonlinear hardware model was mapped from the 13,000 iterations linear model using the adaptation algorithm but without fine-tuning, and the nonlinear hardware model with adaptation and fine-tuning was adapted from a linear model trained on 3,000 iterations, and then fine-tuned on 10,000 iterations.

### Hardware SPICE simulations

To assess circuit performance accuracy, energy consumption and speed, we conducted SPICE array simulations using the TSMC 28 nm PDK within the Cadence Virtuoso environment. All simulations are based on a $64 \times 64$ array, corresponding to the tile size in our architecture (Fig. 3a). To extrapolate the energy and latency for a full attention head with a window size of 1,024, we multiply the per-sub-tile measurements by 16, reflecting the total number of sub-tiles comprising 1 attention head in our architecture. In these simulations, a parasitic wire capacitance of 0.8 fF and a series resistance of 2 Ω per array element are included. Both arrays, one performing $\phi(Q \cdot K^T)$ and the other performing $\phi(S) \cdot V$, are simulated separately, but always in combination with their specific charge-to-pulse circuitry readout circuitry.

### GPU attention latency and energy consumption measurements

To measure the latency and energy on Nvidia RTX 4090, Nvidia H100 and Nvidia Jetson Nano, which are a consumer GPU, a data-center GPU and an embedded application GPU, respectively, we perform 10 runs of 1,024 steps of autoregressive token generation with 12 attention heads using the method FlashAttention-2[51], which optimizes attention computation in GPUs. The energy and latency consumption measurement solely focus on attention computation, and for a fair comparison, the linear projections are not implemented in this experiment as they are also not implemented by our hardware architecture, and the static power measured before inference is subtracted from the power measured during inference. For each run, we measure the latency and the power using the Nvidia-SMI python API, and average them.

### Area estimation

Our floorplan is based on ITO gain cells, an emerging OSFET technology that has enabled low-area gain-cell designs[45]. A two-transistor ITO gain cell occupies an area of 0.14 µm² (approximately 370 nm × 370 nm)[45], allowing for denser memories than CMOS-based gain cells. On the basis of the area results presented in these studies[45,46], we estimate the worst-case area of the proposed 6-transistor cell to be 1 µm², leading to a 19× area reduction compared with gain cells based on CMOS write transistors (our CMOS-based gain-cell layout is presented in Supplementary Fig. 1). The total area of 1 attention head is derived from this single-cell area estimation, as well as the charge-to-pulse circuit layout and the total floorplan incorporating the 16 sub-tiles and digital circuits, providing a precise representation of the space requirements. This structure is designed to be repetitive (vertical dimension in Fig. 3c), allowing multiple attention heads to be efficiently integrated on a single chip. Each attention head receives inputs from the lower digital block, while its outputs are processed by the upper digital block. To facilitate the connection of the bitline outputs of one array (that is, vertical metal lines) to the wordline input of the next array (that is, horizontal metal line), we employ wire tapping, as highlighted in Fig. 3d.

When considering 3D-stacked gain cells, the effective cell area is reported in ref. 45 as $0.14/N$ µm², where $N$ denotes the number of parallel oxide layers. Consequently, a signed gain-cell implementation would occupy $0.28/N$ µm², consisting of 2 gain cells, 1 for the positive part and 1 for the negative part.

## Data availability

The data supporting the figures of this study are publicly available in a figshare repository[63]. Source data for Figs. 1, 2, 4 and 5 are available with this paper. Data for Figs. 1, 2 and 5 were generated through simulations using SPICE. Data for Fig. 4 were produced using evaluations performed in the PyTorch framework. Data for Table 1 were obtained using the Language Model Evaluation Harness toolkit[64].

## Code availability

The Python scripts used for the experiments are available without restriction at https://github.com/NathanLeroux-git/GainCellAttention/, and are archived with a DOI in the Zotero repository[65].

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

## Acknowledgements
This work was supported in part by the Federal Ministry of Education and Research (BMBF, Germany) in the project NEUROTEC II (project number 16ME0398K). We gratefully acknowledge the Gauss Centre for Supercomputing e.V. (www.gauss-centre.eu) for funding this project by providing computing time through the John von Neumann Institute for Computing (NIC) on the GCS Supercomputer JUWELS at Jülich Supercomputing Centre (JSC).

## Author contributions
The study was designed by N.L. and P.-P.M., and supervised by J.P.S. and E.N. The analog circuit system schematic design and electrical simulations were carried out by P.-P.M. C.S. was responsible for the design and layout of all digital blocks, as well as the overall chip floorplanning. S.S. completed the layout of the analog components. Hardware parameter extraction was performed by P.-P.M. Neural network training was conducted by N.L. and neural network evaluation was conducted by N.L. and J.F. All authors contributed to the analysis of the results and writing of the paper.

## Funding

## Competing interests
The authors declare no competing interests.

## Additional information

**Correspondence and requests for materials** should be addressed to Nathan Leroux or Paul-Philipp Manea.

