## [Peer Review file · Nature Computational Science]

Analog In-Memory Computing Attention Mechanism for Fast and Energy-Efficient Large Language Models

Corresponding Author: Dr Nathan Leroux

Version 0:

Decision Letter:

** Please ensure you delete the link to your author homepage in this e-mail if you wish to forward it to your co-authors. **

Dear Dr Leroux,

Your manuscript "Analog In-Memory Computing Attention Mechanism for Fast and Energy-Efficient Large Language Models" has now been seen by 3 referees, whose comments are appended below. You will see that while they find your work of interest, they have raised points that need to be addressed before we can make a decision on publication.

The referees' reports seem to be quite clear. Naturally, we will need you to address **all** of the points raised.

While we ask you to address all of the points raised, the following points need to be substantially worked on:

- All referees raised their concerns on your model's capability in generating to larger models. We suggest you provide more experiments on larger models, as suggested by referees.
- Please improve the comparisons in your study, as suggested by referees.
- Please broaden the background discussion, especially on those works on reducing the computing budget in attention layers.
- Please provide clear justifications on the choice of ReLU attention mechanism as a demo in the study.

Please use the following link to submit your revised manuscript and a point-by-point response to the referees' comments (which should be in a separate document to any cover letter):

Link Redacted

** This url links to your confidential homepage and associated information about manuscripts you may have submitted or be reviewing for us. If you wish to forward this e-mail to co-authors, please delete this link to your homepage first. **

To aid in the review process, we would appreciate it if you could also provide a copy of your manuscript files that indicates your revisions by making use of Track Changes or similar mark-up tools. Please also ensure that all correspondence is marked with your Nature Computational Science reference number in the subject line.

In addition, please make sure to upload a Word Document or LaTeX version of your text, to assist us in the editorial stage.

If you have any issues when updating your Code Ocean capsule during the revision process, please email the Code Ocean support team Cc'ing me.

To improve transparency in authorship, we request that all authors identified as 'corresponding author' on published papers create and link their Open Researcher and Contributor Identifier (ORCID) with their account on the Manuscript Tracking System (MTS), prior to acceptance. ORCID helps the scientific community achieve unambiguous attribution of all scholarly contributions. You can create and link your ORCID from the home page of the MTS by clicking on 'Modify my Springer Nature account'. For more information please visit www.springernature.com/orcid.

We hope to receive your revised paper within three weeks. If you cannot send it within this time, please let us know.

Best regards,

Jie Pan, Ph.D.
Senior Editor
Nature Computational Science

Reviewers comments:

Reviewer #1 (Remarks to the Author):

Leroux et al. introduce a replacement for the softmax attention layer, typically used as one of the key components of a deep transformer architecture, alongside with an efficient analog hardware implementation, and a method for hardware-aware training for these models. Compelling results are shown for smaller-scale language models (GPT2, Wikitext-2, etc.). The presentation of the results is clear and the contribution to the field is interesting. The manuscript falls short in the discussion of related literature and in assessing the impact of the proposed ReLU attention mechanism. For the latter, additional experiments would be required to support the claims made throughout the manuscript.

Major issues:

To achieve the gain in speed and performance, the architecture replaces the softmax layer, that is most typically used in transformer architectures, with a ReLU nonlinearity. The search for a replacement for the self-attention layers with linear time complexity is an interesting research topic that has sparked the development of new ML architectures. For example, state space models have made quite impressive advancements in recent years, showing performances comparable to even the largest transformer-based LLMs using linear-time-complexity attention layers (see e.g. Gu and Dao's MAMBA for a model of the newest generation). However, these models use some very well-designed additional constructions to mix input tokens across the time domain, and it is well established that simply using a purely linear (or ReLU) attention doesn't provide the same computational benefits as a softmax that perfectly correlates all tokens across time. In light of these results, the manuscript seems to overstate or over-simplify the matter a bit. At several places, the authors claim that the model has favorable scaling properties, but little evidence is provided. GPT2 is not a state-of-the-art transformer, and earlier results have shown that scaling issues may arise for linear/ReLU attention with billion-parameter models. But it is these very large models that show the most interesting properties, such as in-context learning, and a hardware architecture that survives the test of time should be able to reach this scale. Therefore, it would be crucial to provide scaling analyses with models of contemporary size (GPT3-4). The relatively low performance reported on LAMBADA and more complex tasks may point in the direction that scaling is in fact an issue here. These experiments could be done in software to prove that the proposed architecture can in principle reach modern model sizes. In addition, the literature on linear attention and more modern alternatives like MAMBA, S5, ..., should be discussed in some detail to contrast the proposed approach against existing alternative approaches.

Even though, a large body of literature focuses on reducing the compute footprint of attention layer, it is well established that the biggest chunk of the compute budget is consumed by the MLP layers. The authors should discuss the efforts that have been made to speed up also that important part of transformer architectures (see e.g. [2] for a recent survey on the algorithmic side). Also, prior work to discuss alternate approaches to improve the efficiency of large-scale AI models should be discussed in more depth (see e.g. [3] for a recent survey).

Detailed comments:

Page 10: "These results suggest no apparent limitations for NLP tasks using our hardware attention." The results are promising, but scaling to large model sizes might be a major issue here, since ReLU units typically show less favorable scaling properties (see major comment above).

Page 11: 0.5 mm² is a non-trivial die size, given the 12 layers and 12 heads, so total of 144 self-attention layers for GPT2, thus totaling to 72 mm² across all layers/heads. Larger, contemporary models, like GPT3/4 would potentially require significantly larger chip areas or additional memory to buffer embeddings. The authors should discuss in detail how this would affect efficiency, and the 0.5 mm² should be put in perspective of the chip area vs. energy efficiency trade-off.

Page 12f: "Our architecture can benefit from OSFET transistors that enable dense 3D integration". 3D Stacking is a very interesting topic, but is often infeasible due to challenges in thermal management. An estimate on the potential for 3D stacking should be given, taking together the measured energy consumption and other potential heat sources in the architecture.

Page 14: "... and therefore we conclude that the choice of decay factor of 1.6×10^{-4} is very conservative." This analysis should be expanded a bit. As discussed above, MLPs are completely ignored here. Also, the 12 layer GPT2 is far from a modern model size, so the delays could be significantly longer than this "conservative" estimate. How will deeper architectures, and resulting longer delays, affect the specs of gain cell memories, e.g. with respect to die size/area

requirements?

[1] Gu and Dao, MAMBA, <https://arxiv.org/pdf/2312.00752>

[2] XU et al. 2024. <https://xumengwei.github.io/files/CSUR24-efficientllm.pdf>

[3] Vogginger et al. 2024. <https://arxiv.org/pdf/2402.02521>

Reviewer #1 (Remarks on code availability):

The code is an adaptation of OpenAI's GPT2 implementation, with custom code for attention layers and training procedure. The README file seems complete. The script for training the model is one very long block of code, that would benefit from additional comments and being structured a bit more.

Reviewer #2 (Remarks to the Author):

This work presents a self-attention in-memory computing architecture based on emerging charge-based memory devices known as gain cells. These gain cells enable efficient token storage during sequence generation and facilitate parallel analog dot-product computations required for self-attention. However, the analog characteristics of the gain cell circuits introduce non-idealities and constraints, which prevent the direct application of pre-trained models. To overcome this challenge, the authors design an initialization algorithm that achieves text processing performance comparable to GPT-2 without the need for training from scratch. The proposed architecture reduces attention latency and energy consumption by up to two and five orders of magnitude, respectively, compared to GPUs.

This work focuses on reducing the computational overhead of self attention through algorithmic analysis and circuit design, providing a relatively comprehensive approach. However, I have the following questions and comments.

Comments:

1. One of the improvements to the attention mechanism proposed in this work is the use of ReLU, but the authors do not provide a clear explanation for this choice. Is it motivated by ease of implementation in the circuit design? To the best of my knowledge, applying an element-wise nonlinearity after the QK dot product offers limited advantages, as it neither facilitates the decomposition into linear attention nor preserves the benefits of the softmax function.
2. Flash Attention is an optimized computational method for the self-attention mechanism on GPU hardware. Its key insight is that, in most cases, the bottleneck of the self-attention mechanism lies in memory limitations. As a memory-compute integrated design for linear attention, I do not see the advantages of this approach. Additionally, the authors should compare the proposed method with Flash Attention in terms of computational latency and energy consumption.
3. What does the "public software model" in Table 1 refer to? If it refers to GPT-2, the reported performance metrics are inconsistent with the original paper. For instance, the GPT-2 117M model achieves an accuracy of 45.99 and a perplexity of 35.13 on LAMBADA as stated in the original paper, whereas the authors report results of 32.56 and 40.06, respectively.
4. Using sliding window attention may lead to suboptimal performance. I strongly recommend that the authors discuss the use of current mainstream block-based self-attention methods (such as the chunk-wise approach in Flash Attention) to optimize the proposed method.
5. The authors claim to optimize large language models; however, the experiments are conducted using a GPT-2 model with only 124 million parameters (0.124B). Given the scaling laws and potential outliers, I am concerned that the proposed architecture may suffer significant performance degradation when scaled to models with billions of parameters. I recommend that the authors validate their method on larger models, such as LLaMA2-7B, to better assess its scalability and effectiveness.

Reviewer #2 (Remarks on code availability):

Codes are nice for readers to reproduce, and the detailed readme file is provided.

Reviewer #2 (Remarks on figshare data availability):

These data are in agreement with the claims of the Article.

Reviewer #3 (Remarks to the Author):

This paper proposes a custom self-attention in-memory computing (IMC) architecture to address the energy and latency bottlenecks in large language models (LLMs). Specifically, the authors leverage capacitor-based analog gain cells as memory and computation units to store key-value (KV) projections and perform parallel dot-product computations central to the self-attention mechanism. A hardware-aware adaptation algorithm enables pre-trained language models (e.g., GPT-2) to operate effectively on this non-ideal analog hardware, without requiring retraining from scratch. However, the architecture's reliance on analog components introduces trade-offs in accuracy and generalizability, while the justified approach on the

attention mechanism limits its broader applicability. The reliance on simulations rather than physical implementations further diminishes the immediate impact of the work. As a result, I cannot recommend the manuscript for publication, at least not in the present form.

1) It should be noted that the GPT-2 model selected by the authors is described as having 117 million parameters in the original paper [1], which is inconsistent with the description provided by the authors in section 4.2 of this paper. Additionally, in the original paper, the 12-layer GPT-2 has a perplexity (PPL) of 35.13 and an accuracy (ACC) of 45.99 on LAMBADA, while the perplexity on WikiText2 is 29.41. These numbers differ significantly from the data presented by the authors in TABLE 1. The authors have seemingly diminished the performance of the baseline considerably. Compared to the original performance, there is a considerable gap in task performance with the proposed method in this paper.

2) Due to hardware constraints, the authors have made some adjustments to the transformer algorithm, replacing the softmax function after $Q \cdot K^T$ with ReLU. In the original transformer, the softmax function not only normalizes the attention between tokens but also introduces competition between the attentions of different tokens. By replacing it with ReLU, the authors have only removed negative attention values, which leads to two results: i) The possibility of extremely large values; ii) Each attention value becomes an independent attention. The authors conducted experiments only on the 12-layer GPT-2, and according to the original data of GPT-2, there is still a performance gap with the proposed method. This gap is expected to widen when dealing with larger models.

3) The authors propose using sliding window attention to replace causal attention, which also achieves the masking functionality in decoder-only LLMs. However, it also results in the loss of a significant amount of needed attention, such as the attention between Q3 and K0 in fig 2.a. The impact of such lost attention will accumulate with larger models, potentially having a huge effect on task performance when dealing with larger models.

4) The authors mention that resistive losses in interconnects, known as IR drop issues, can cause reduced accuracy in large-scale analog crossbar arrays. They mitigate this by limiting the size of their gain cell arrays. However, how does this size limitation impact the scalability of the proposed architecture? This could potentially limit the system's ability to handle larger models or more complex tasks, which could affect its broad applicability and usefulness in real-world applications. And it could also introduce additional complexity and potential overheads, affecting the overall efficiency of the architecture.

5) The authors' architecture utilizes storage capacitors in its design, which are known to have retention issue leading to stored value decay over time. How does this leakage impact the long-term performance and reliability of the proposed system? While this is a less critical limitation in short-sequence processing, if not appropriately managed, limited retention could lead to a gradual degradation in system performance, potentially affecting the accuracy of computations and limiting the system's effectiveness in long-term sequence processing or scenarios requiring extended operations.

Reference:

[1]Radford, Alec, et al. "Language models are unsupervised multitask learners." OpenAI blog 1.8 (2019): 9.

Version 1:

Decision Letter:

Our ref: NATCOMPUTSCI-24-2552A

16th June 2025

Dear Dr. Leroux,

Thank you for submitting your revised manuscript "Analog In-Memory Computing Attention Mechanism for Fast and Energy-Efficient Large Language Models" (NATCOMPUTSCI-24-2552A). It has now been seen by the original referees and their comments are below. The reviewers find that the paper has improved in revision, and therefore we'll be happy in principle to publish it in Nature Computational Science, pending minor revisions to satisfy the referees' final requests and to comply with our editorial and formatting guidelines.

We are now performing detailed checks on your paper and will send you a checklist detailing our editorial and formatting requirements in about two weeks. Please do not upload the final materials and make any revisions until you receive this additional information from us.

TRANSPARENT PEER REVIEW

Nature Computational Science offers a transparent peer review option for original research manuscripts. We encourage increased transparency in peer review by publishing the reviewer comments, author rebuttal letters and editorial decision letters if the authors agree. Such peer review material is made available as a supplementary peer review file. **Please remember to choose, using the manuscript system, whether or not you want to participate in transparent peer review.**

Please note: we allow redactions to authors' rebuttal and reviewer comments in the interest of confidentiality. If you are concerned about the release of confidential data, please let us know specifically what information you would like to have removed. Please note that we cannot incorporate redactions for any other reasons. Reviewer names will be published in the peer review files if the reviewer signed the comments to authors, or if reviewers explicitly agree to release their name. For

more information, please refer to our [FAQ page](https://www.nature.com/documents/nr-transparent-peer-review.pdf).

Thank you again for your interest in Nature Computational Science. Please do not hesitate to contact me if you have any questions.

Sincerely,

Jie Pan, Ph.D.
Senior Editor
Nature Computational Science

ORCID

Reviewer #1 (Remarks to the Author):

I want to thank the authors for addressing some of my concerns. The manuscript was improved, but some major issues remain:

The new results on GPT-2-XL are not very convincing and the interpretation flawed. For example, the accuracy on ARC-E dataset drops from 58.29 to 53.79 with the proposed HW model, while the accuracies were quite similar for the GPT-2 model. Hence, there is a significant drop of 4.5, but still the authors conclude "... the gap with its software counterparts is not larger than for GPT-2". However, I am not convinced that the proposed simplified replacement of the attention mechanisms can be scaled to state-of-the-art model sizes (other than the authors claim "This demonstrates that increasing model size with our hardware-based attention mechanism leads to improved accuracy"). The manuscript does not provide a detailed mathematical analysis of the impact of the proposed method. In that same paragraph on page 11, the authors claim "We attribute the remaining performance gap to quantization effects and the use of HardSigmoid activation, which clamp values and therefore can cause vanishing gradients when inputs fall outside the activation range", without a justification. It would be necessary to conduct additional experiments, and rigorous theoretical modelling of the proposed approximations to quantify the impact. In addition, I would recommend testing the model on at least one additional model size of > 3 B parameters. Such an experiment would more clearly show the trend and justify the claims made throughout the paper. Without such a justification, the many claims about the scalability of their method need to be dropped.

In their added text on page 2 the authors claim: "However, because of their parallelizable training, to this day Transformers are still favoured over models such as Mamba [13]", but that is wrong. Mamba is highly optimized for efficient use of hardware and has demonstrates excellent scaling properties. In fact, the parallel scan algorithm that can be efficiently executed in parallel on GPU hardware was one of the main innovations underlying MAMBA (also other SSM architectures share this property). The reviewer acknowledges that transformers are still heavily used despite their unfavourable N^2 time complexity, but parallelizable training is not the issue here. Please rephrase this paragraph to resolve this very misleading statement.

The system level implementation outlined in 5.1 is somewhat contrived and incomplete. The proposed method uses an external hardware block that implements the required intermediate MLP layers, while only self-attention is done on the chip. The main advantage of 3D stacking is the reduction of wiring lengths. However, this advantage is completely lost here, since attention layers are simply stacked on top of each other, while the required intermediate MLP parts are left externally. Also, the authors show in their estimation of the energy budget in section 5.2 that only a single of these layers can be activated at a time. This seems quite wasteful, in light of the 3D stacking. If the energy budget is prohibitive of parallel pipelining (e.g. batch-parallel execution of the multiple prompts), this seems to suggest that the architecture is ill-suited for 3D stacking altogether.

Reviewer #1 (Remarks on code availability):

The added comments are helpful, thanks.

Reviewer #2 (Remarks to the Author):

Thank you for addressing my previous comments; they have resolved most of my concerns. I still have reservations about Comment 2—the details of the latency and energy-consumption comparisons on GPUs. I believe the authors should provide (1) a broader range of GPU models, (2) a larger context window (e.g., 10–2048 tokens), and (3) a wider set of model baselines. Because latency and energy measurements are independent of model accuracy, the authors could also evaluate larger LLaMA variants. Strengthening these comparisons would better highlight the contribution of the work.

The experimental evaluation should be performed on mainstream NVIDIA GPUs from the Ampere and Hopper generations that provide native support for Flash-Attention and Flash-Attention V2 (e.g., the A100, H100, and H200). We are, moreover,

keen to observe the method's behaviour on alternative accelerators, such as Huawei Ascend 910C and AMD Instinct MI300X. The benchmark models ought to span a wide parameter spectrum approximately 1 B to 70 B parameters. Suitable model families include Qwen, LLaMA, and DeepSeek, and sequence lengths should range from 128 to 1 024 tokens. We strongly encourage the authors to conduct the above experiments to furnish a more comprehensive comparison of their approach against prevailing parallel-computing architectures.

Reviewer #2 (Remarks on code availability):

The code provides a README file with enough instructions, and readers can easily run it. I checked the code but didn't run it.

Reviewer #2 (Remarks on figshare data availability):

They are consistent with the results in the article.

Reviewer #3 (Remarks to the Author):

I appreciate the detailed response from the authors and the substantial work to improve the quality of this work. I'm pleased to see that my previous concerns have been addressed in this revision. The authors have carefully revised the manuscript accordingly. I would like to recommend this paper to be accepted for publication.

Version 2:

Decision Letter:

Dear Dr Leroux,

We are pleased to inform you that your Article "Analog In-Memory Computing Attention Mechanism for Fast and Energy-Efficient Large Language Models" has now been accepted for publication in Nature Computational Science.

Once your manuscript is typeset, you will receive an email with a link to choose the appropriate publishing options for your paper and our Author Services team will be in touch regarding any additional information that may be required.

Authors may need to take specific actions to achieve compliance with funder and institutional open access mandates. If your research is supported by a funder that requires immediate open access (e.g. according to [Plan S principles](https://www.springernature.com/gp/open-science/plan-s-compliance) or the [NIH public access policy](https://www.springernature.com/gp/open-science/us-federal-agency-compliance)) then you should select the gold OA route, and we will direct you to the compliant route where possible. Because authors warrant under our subscription licensing terms that they haven't committed to licensing any version of their article under a licence inconsistent with the terms of our agreement – including the applicable embargo period – publication under the subscription model isn't suitable for authors whose funders require no embargo.

Acceptance of your manuscript is conditional on all authors' agreement with our publication policies (see <https://www.nature.com/natcomputsci/for-authors>). In particular your manuscript must not be published elsewhere and there must be no announcement of the work to any media outlet until the publication date (the day on which it is uploaded onto our web site).

Before your manuscript is typeset, we will edit the text to ensure it is intelligible to our wide readership and conforms to house style. We look particularly carefully at the titles of all papers to ensure that they are relatively brief and understandable.

Once your manuscript is typeset, you will receive a link to your electronic proof via email with a request to make any corrections within 48 hours. If, when you receive your proof, you cannot meet this deadline, please inform us at rjsproduction@springernature.com immediately.

If you have queries at any point during the production process then please contact the production team at rjsproduction@springernature.com.

We welcome the submission of potential cover material (including a short caption of around 40 words) related to your manuscript; suggestions should be sent to Nature Computational Science as electronic files (the image should be 300 dpi at 210 x 297 mm in either TIFF or JPEG format). We also welcome suggestions for the Hero Image, which appears at the top of our [home page](http://www.nature.com/natcomputsci); these should be 72 dpi at 1400 x 400 pixels in JPEG format. Please note that such pictures should be selected more for their aesthetic appeal than for their scientific content, and that colour images work better than black and white or grayscale images. Please do not try to design a cover with the Nature Computational Science logo etc., and please do not submit composites of images related to your work. I am sure you will understand that we cannot make any promise as to whether any of your suggestions might be selected for the cover of the journal.

Best regards,

Jie Pan, Ph.D.
Senior Editor
Nature Computational Science

P.S. Click on the following link if you would like to recommend Nature Computational Science to your librarian: <https://www.springernature.com/gp/librarians/recommend-to-your-library>

** Visit the Springer Nature Editorial and Publishing website at <http://editorial-jobs.springernature.com> for more information about our career opportunities. If you have any questions please click [here](mailto:editorial.publishing.jobs@springernature.com). **

Reviewer #1 (Remarks to the Author):

Leroux et al. introduce a replacement for the softmax attention layer, typically used as one of the key components of a deep transformer architecture, alongside with an efficient analog hardware implementation, and a method for hardware-aware training for these models. Compelling results are shown for smaller-scale language models (GPT2, Wikitext-2, etc.). The presentation of the results is clear and the contribution to the field is interesting. The manuscript falls short in the discussion of related literature and in assessing the impact of the proposed ReLU attention mechanism. For the latter, additional experiments would be required to support the claims made throughout the manuscript.

Major issues:

To achieve the gain in speed and performance, the architecture replaces the softmax layer, that is most typically used in transformer architectures, with a ReLU nonlinearity. The search for a replacement for the self-attention layers with linear time complexity is an interesting research topic that has sparked the development of new ML architectures. For example, state space models have made quite impressive advancements in recent years, showing performances comparable to even the largest transformer-based LLMs using linear-time-complexity attention layers (see e.g. Gu and Dao's MAMBA for a model of the newest generation). However, these models use some very well-designed additional constructions to mix input tokens across the time domain, and it is well established that simply using a purely linear (or ReLU) attention doesn't provide the same computational benefits as a softmax that perfectly correlates all tokens across time. In light of these results, the manuscript seems to overstate or over-simplify the matter a bit. At several places, the authors claim that the model has favorable scaling properties, but little evidence is provided. GPT2 is not a state-of-the-art transformer, and earlier results have shown that scaling issues may arise for linear/ReLU attention with billion-parameter models. But it is these very large models that show the most interesting properties, such as in-context learning, and a hardware architecture that survives the test of time should be able to reach this scale. Therefore, it would be crucial to provide scaling analyses with models of contemporary size (GPT3-4). The relatively low performance reported on LAMBADA and more complex tasks may point in the direction that scaling is in fact an issue here. These experiments could be done in software to prove that the proposed architecture can in principle reach modern model sizes. In addition, the literature on linear attention and more modern alternatives like MAMBA, S5, ..., should be discussed in some detail to contrast the proposed approach against existing alternative approaches.

Even though, a large body of literature focuses on reducing the compute footprint of attention layer, it is well established that the biggest chunk of the compute budget is consumed by the MLP layers. The authors should discuss the efforts that have been made to speed up also that important part of transformer architectures (see e.g. [2] for a recent survey on the algorithmic side). Also, prior work to discuss alternate approaches to improve the efficiency of large-scale AI models should be discussed in more depth (see e.g. [3] for a recent survey).

[1.A] Gu and Dao, MAMBA, <https://arxiv.org/pdf/2312.00752>

[2] XU et al. 2024. <https://xumengwei.github.io/files/CSUR24-efficientllm.pdf>

[3] Vogginger et al. 2024. <https://arxiv.org/pdf/2402.02521>

Answer:

We thank the reviewer for these insightful comments. We would like to clarify that despite our system not having softmax, our work does not fit in the category of linear Transformers such as¹, and not either in the category of state-space models. These different linear-time complexity alternatives aim to compress the past states in a fixed size memory. On the other hand, our self-attention is similar to conventional Transformers, as the two dot-products of our system compare explicitly the present query with all keys and values from different time steps. Despite the normalization of softmax being a useful feature during training to avoid under or over-flow, it is not a necessary step to build correlations between tokens, which is already done with dot-products. Moreover, even though softmax is still the most commonly used nonlinearity, it has been shown that the normalization in softmax can as well bring issues such as attention sink which focuses attention on non informative tokens², or information loss in sliding window attention³.

We added a new paragraph in the introduction to discuss the innovation in linear-time complexity alternatives such as S5 or Mamba, as well as other algorithmic improvements for Transformers: *“To mitigate this bottleneck, a wide body of literature explores resource-efficient algorithms [9]. Alternative architectures to Transformers with linear time complexity are developed to improve long sequence processing efficiency [10–13]. However, because of their parallelizable training, to this day Transformers are still favored over models such as Mamba [13]. Alternatively different methods are developed to reduce the memory requirements of KV-caching through token pruning [14, 15], latent KV-cache compression [16, 17], low-rank approximations [18, 19], or by reusing the same KV-cache pairs across multiple heads (Grouped-Query Attention) [20, 21].”*

In⁴, the softmax is replaced by an element-wise sigmoid function. On natural language processing and image processing tasks, their model achieves accuracies similar to those of softmax-based attention on 7 billion parameters models. Since our activation function saturates (see figure 1(g)), it is not a ReLU and is more similar to HardSigmoid. We apologize for the confusion on terminology, and replace the name ReLU with the more suited HardSigmoid name in the text. We also added a sentence in the introduction to explain the possibility of using Sigmoid-based attention: *“In particular, Sigmoid-based attention has been shown to match Softmax-based attention on models up to 7 billion parameters large [48]. Recent works show that in the case of Sliding Window Attention [49], the normalization of Softmax leads to vanishing memory while Sigmoid-based attention can lead to better information [50, 51].”*

We acknowledge the importance of scaling to larger models. For the revised article, we trained GPT-2-XL, which has 1,5 billion parameters (please see later answer), which is 10 times larger than the previous model. However, due to computational resources, we cannot train GPT3 nor GPT4, which has above 1 trillion parameters and therefore 10.000 larger than what we did.

MLPs are indeed a major contribution to the overall compute budget. However, as pointed out in⁵ and⁶, attention is still a very meaningful part of the compute budget, especially when for large sequence lengths. Furthermore, In-Memory Computing methods can already significantly reduce the MLPs cost⁷. Therefore, in a hardware implementation with In-Memory Computing-based MLPs but with conventional digital logic-based attention units, the attention would be the next bottleneck, which we address here. In the introduction, we added a new discussion about existing hardware-efficient implementations and integrated the proposed survey: *“While these algorithmic strategies reduce computational and memory overhead, achieving further energy efficiency increasingly depends on hardware innovation. [...] However, a full optimization of Transformers’ inference also requires addressing the attention mechanism, which contributes significantly to the overall computational cost [9, 25].”*

References:

¹ Katharopoulos, A., Vyas, A., Pappas, N. & Fleuret, F. Transformers are RNNs: Fast Autoregressive Transformers with Linear Attention. Preprint at <http://arxiv.org/abs/2006.16236> (2020).

² Gu, X. et al. When Attention Sink Emerges in Language Models: An Empirical View. Preprint at <https://doi.org/10.48550/arXiv.2410.10781> (2025).

³ Fu, Z. et al. Sliding Window Attention Training for Efficient Large Language Models. Preprint at <https://doi.org/10.48550/arXiv.2502.18845> (2025).

⁴ Ramapuram, J. et al. Theory, Analysis, and Best Practices for Sigmoid Self-Attention. Preprint at <https://doi.org/10.48550/arXiv.2409.04431> (2025).

⁵ Wolters, C., Yang, X., Schlichtmann, U. & Suzumura, T. Memory Is All You Need: An Overview of Compute-in-Memory Architectures for Accelerating Large Language Model Inference. Preprint at <https://doi.org/10.48550/arXiv.2406.08413> (2024).

⁶ Laguna, A. F. et al. Hardware-Software Co-Design of an In-Memory Transformer Network Accelerator. *Frontiers in Electronics* 3, (2022).

⁷ Sebastian, A., Le Gallo, M., Khaddam-Aljameh, R. & Eleftheriou, E. Memory devices and applications for in-memory computing. *Nat. Nanotechnol.* 15, 529–544 (2020).

Detailed comments:

Page 10: “These results suggest no apparent limitations for NLP tasks using our hardware attention.” The results are promising, but scaling to large model sizes might be a major issue here, since ReLU units typically show less favorable scaling properties (see major comment above).

Answer:

In response, we trained GPT-2-XL which has 1,5 billion parameters. In the section results 2.5, we added the paragraph: *“To evaluate scalability on larger models, we applied the same training methodology to GPT-2-XL, which contains 1.5 billion parameters. While its performance is slightly lower than the software-only baseline, it significantly outperforms the smaller GPT-2 model. Moreover, the gap with its software counterparts is not larger than for GPT-2. This demonstrates that increasing model size with our hardware-based attention mechanism leads to improved accuracy. We attribute the remaining performance gap to quantization effects and the use of HardSigmoid activation, which clamps values and therefore can cause vanishing gradients when inputs fall outside the activation range.”* and updated the corresponding benchmark table.

We acknowledge that our statement “These results suggest no apparent limitations for NLP tasks using our hardware attention” was over-simplified and have now deleted it. Instead, in the discussion we added the sentences: *“Nonetheless, our larger network slightly underperforms the baseline, and therefore deeper neural network training will require further methods to mitigate the vanishing gradient issue due to clamping values. This slight performance gap should still be put in perspective with the reduced energy consumption.”*

Page 11: 0.5 mm² is a non-trivial die size, given the 12 layers and 12 heads, so total of 144 self-attention layers for GPT2, thus totaling to 72 mm² across all layers/heads. Larger, contemporary models, like GPT3/4 would potentially require significantly larger chip areas or additional memory to buffer embeddings. The authors should discuss in detail how this would affect efficiency, and the 0.5 mm² should be put in perspective of the chip area vs. energy efficiency trade-off.

Answer:

We thank the reviewer for highlighting the importance of contextualizing chip area in terms of efficiency and scalability, especially in light of large-scale models.

We would like to clarify that our Gain Cells crossbar arrays are KV-cache buffers, and therefore no additional memory is required for attention. We added a sentence to clarify this point in the introduction: *“As a result, gain cell crossbar arrays simultaneously serve to store the KV-cache and to perform attention computation.”*

Regarding die area considerations for larger models, we want to emphasize that the requirements for KV-cache memory do not scale proportionally with the number of model parameters. To illustrate this, we analyzed the ratio of KV pairs to parameter count across various GPT-3 model sizes, as shown in this figure:

Evolution of the ratio between KV-cache and number of parameters for multiple increasingly large GPT3 models. From [<https://doi.org/10.48550/arXiv.2005.14165>].

This example suggests that the required KV-cache size, and therefore the associated hardware area scales sub-linearly with the model size. We added a sentence in the discussion section: *"Moreover, the KV-cache size grows modestly compared to the overall models' parameters count [18–20, 58]. Our system could therefore be applied to larger networks with a moderate area footprint."*

Additionally, it is important to note that GPT-2 and GPT-3 employ standard (vanilla) attention mechanisms without any KV-cache optimizations. A variety of techniques have since been proposed to further reduce KV-cache memory size requirements. We added a paragraph to describe these methods in the introduction: *"Alternatively different methods are developed to reduce the memory requirements of KV-caching through token pruning [14, 15], latent KV-cache compression [16, 17], low-rank approximations [18, 19], or by reusing, the same KV-cache pairs across multiple heads (Grouped-Query Attention) [20, 21]."*

Furthermore, we emphasize that the current area estimation for the gain cell of $1\mu\text{m}^2$ represents a worst-case scenario. More compact implementations have already been demonstrated, as discussed in Section~4.6 of the initial version of our manuscript: *"Our floorplan is based on ITO gain cells, an emerging OSFET technology that has enabled low area gain cell designs [56]. A two transistor ITO gain cell occupies an area of $0.14\ \mu\text{m}^2$ ($\approx 370\ \text{nm} \times 370\ \text{nm}$) [56], allowing for denser memories than CMOS-based gain cells."* Total area requirement estimates are provided in Section~2.8: *"However, other works have demonstrated significantly smaller gain-cell dimensions [56]. Based on this, and following the methodology outlined in Section 4.6, we estimate that the area of the gain-cell crossbars required for the entire GPT-2 attention head KV-cache is approximately $15.7 \times 10^{-3}\ \text{mm}^2$, excluding digital control circuitry."*

Effective Area can be further decreased utilizing the 3D capabilities of Oxide Semiconductor Field Effect Transistor (OSFET). We therefore added in the introduction the sentence: *"can be manufactured with very small feature sizes, achieving higher density than SRAM, and also support 3D integration, which can further reduce effective area requirements [36–41]."* In the supplementary material, we added a new section *"Mapping Transformer Operations to 3D OSFET In-Memory Computing Layers"*, where we provide an insight into the operation of a Transformer accelerator utilizing 3D capabilities while taking hardware constraints into

account. In the results section, we added the paragraph: “*In supplementary material 5, we demonstrate that multiple attention heads can be executed using parallel tiles on-chip and stacked in 3D with multiple layers, sharing peripheral and digital logic, as illustrated in supplementary material Fig. 7.*” We provide estimates based on reported literature, where we describe the methodology in: “*When considering 3D-stacked gain cells, the effective cell area is reported in [56] as $0.14/N \mu\text{m}^2$, where N denotes the number of parallel oxide layers. Consequently, a signed gain-cell implementation would occupy $0.28/N \mu\text{m}^2$.*” The total area versus number of stacks is described in the results section: “*Based on [56], we estimate the total area required for a GPT attention head KV-cache, excluding digital control, to be $36.7 N \times 10^{-3} \text{mm}^2$, where N denotes the number of vertical stacks. The resulting area is:*

- $36.7 \times 10^{-3} \text{mm}^2$ for $N = 1$,
- $9.2 \times 10^{-3} \text{mm}^2$ for $N = 4$,
- $4.6 \times 10^{-3} \text{mm}^2$ for $N = 8$,
- $3.1 \times 10^{-3} \text{mm}^2$ for $N = 12$.”

Page 12f: “Our architecture can benefit from OSFET transistors that enable dense 3D integration”. 3D Stacking is a very interesting topic, but it is often infeasible due to challenges in thermal management. An estimate on the potential for 3D stacking should be given, taking together the measured energy consumption and other potential heat sources in the architecture.

Answer:

We thank the reviewer for this insight. To study the potential for 3D stacking, we added a new section 5 (supplemental material) *Mapping Transformer Operations to 3D OSFET In-Memory Computing Layers* in the supplementary materials. In this section, we first propose a design for 3D stacking (see Figure). Our design maps different Transformer attention layers onto different stacks. This design has two main advantages:

- Part of the CMOS hardware can be reused for different Transformer layers.
- Only one layer of the 3D stack is active at a given time, therefore mitigating thermal stress.

By considering the sum of the power dissipation of the CMOS hardware and the power dissipation of the gain cells arrays, we estimate the power density of our architecture to be 37W/cm^2 . It is important to note that this corresponds to the power dissipation during attention inference. However, in practice the hardware for the attention mechanism will not be continuously active since there will be other sources of delay in the full architecture, such as computing the MLPs. The delays between attention inference will give additional time for hardware thermal relaxation.

For comparison, we include the thermal analysis in 3D commercial High Bandwidth Memories (DRAM). The recent analysis in¹ shows that a 3D HBM architecture can function reliably with 40W/cm^2 per stack, with twelve stacks.

In conclusion, we estimate that 3D stacking gain cells arrays could potentially reduce area footprint by one order of magnitude for KV-cache storing. Moreover, because the components are only used sparsely in time, and with a reasonable power dissipation density, we believe that 3D stacked gain cells-based attention mechanisms could be operated without critical

thermal buildup. The above analysis has been added to the manuscript supplementary material.

References:

¹ Son, K. et al. Thermal Analysis of High Bandwidth Memory (HBM)-GPU Module considering Power Consumption. in 2023 IEEE Electrical Design of Advanced Packaging and Systems (EDAPS) 1–3 (2023). doi:10.1109/EDAPS58880.2023.10468315.

Fig. 7 (a) Simplified schematic of a Transformer architecture, illustrating 3 layers, each with 3 parallel attention heads. (b) Proposed system-level architecture mapping Transformer attention heads onto hardware using 3D-stacked OSFET-based In-Memory Computing arrays. Each OSFET stack stores the keys (K) and values (V) corresponding to a different Transformer layer. A single underlying silicon CMOS chip performs non-linear operations, control, and readout.

Page 14: "... and therefore we conclude that the choice of decay factor of 1.6×10^{-4} is very conservative." This analysis should be expanded a bit. As discussed above, MLPs are completely ignored here. Also, the 12 layer GPT2 is far from a modern model size, so the delays could be significantly longer than this "conservative" estimate. How will deeper architectures, and resulting longer delays, affect the specs of gain cell memories, e.g. with respect to die size/area requirements?

Answer:

We thank the reviewer for giving the opportunity to further study this aspect. In the methods section 4.1, we added the sentence: "*In a full Transformer implementation, the latency per layer δt = will be higher than 65 ns as it will also include latency from other modules, such as Feedforward Neural Networks (FNNs).*"

Our claim that this value was conservative was based on the fact that in the literature, Gain Cells retention time have been reported up to 4.5 hours in¹, which is much longer than the 5 ms seconds retention time we considered in this work.

However, the decay factor, which is the ratio between latency and retention time, depends on both the type of Gain Cell and on the latency of the full architecture, which is hard to determine at this stage. Therefore, to study the effect of decay in multiple configurations, we sweep the value of the decay factor for our trained hardware model. The results are shown in the new figure 6 in the supplementary material, and commented in the new section 4 of the supplementary material. Results show that the accuracy starts decreasing when the decay factor is larger than 10^{-3} . It means that if the retention time of the Gain Cells is 10 s, the full architecture latency should not exceed 10 ms (in comparison, the latency per token reported on a GPU H200 on LLAMA 2 70B was 0.1 ms^2). Additionally, training with larger decay could further mitigate negative issues by generalizing to larger decay.

We also added a mathematical analysis in the supplementary material which shows the similarity of the decay with the positional embedding method called ALiBi³. ALiBi is a method used to introduce relative positional information in long sequences. We compare the decay amplitude in our system with the one in ALiBi, and show that ALiBi decay is even larger, while ALiBi allows generalization to larger sequences. This study shows that even larger decays than the one in our system can have a positive effect.

References:

¹ Belmonte, A. et al. Lowest IOFF $< 3 \times 10^{-21} \text{ A}/\mu\text{m}$ in capacitorless DRAM achieved by Reactive Ion Etch of IGZO-TFT. in 2023 IEEE Symposium on VLSI Technology and Circuits (VLSI Technology and Circuits) 1–2 (2023). doi:10.23919/VLSITechnologyandCir57934.2023.10185398.

² NVIDIA Blackwell Platform Sets New LLM Inference Records in MLPerf Inference v4.1. NVIDIA Technical Blog <https://developer.nvidia.com/blog/nvidia-blackwell-platform-sets-new-llm-inference-records-in-mlperf-inference-v4-1/> (2024).

³ Press, O., Smith, N. A. & Lewis, M. Train Short, Test Long: Attention with Linear Biases Enables Input Length Extrapolation. Preprint at <https://doi.org/10.48550/arXiv.2108.12409> (2022).

Reviewer #1 (Remarks on code availability):

The code is an adaptation of OpenAI's GPT2 implementation, with custom code for attention layers and training procedure. The README file seems complete. The script for training the model is one very long block of code, that would benefit from additional comments and being structured a bit more.

Answer:

We thank the reviewer for the comment. To improve readability, we have split the main into different functions and added more comments, as requested.

Reviewer #2 (Remarks to the Author):

This work presents a self-attention in-memory computing architecture based on emerging charge-based memory devices known as gain cells. These gain cells enable efficient token storage during sequence generation and facilitate parallel analog dot-product computations required for self-attention. However, the analog characteristics of the gain cell circuits introduce non-idealities and constraints, which prevent the direct application of pre-trained models. To overcome this challenge, the authors design an initialization algorithm that achieves text processing performance comparable to GPT-2 without the need for training from scratch. The proposed architecture reduces attention latency and energy consumption by up to two and five orders of magnitude, respectively, compared to GPUs.

This work focuses on reducing the computational overhead of self attention through algorithmic analysis and circuit design, providing a relatively comprehensive approach. However, I have the following questions and comments.

Comments:

1. One of the improvements to the attention mechanism proposed in this work is the use of ReLU, but the authors do not provide a clear explanation for this choice. Is it motivated by ease of implementation in the circuit design? To the best of my knowledge, applying an element-wise nonlinearity after the QK dot product offers limited advantages, as it neither facilitates the decomposition into linear attention nor preserves the benefits of the softmax function.

Answer:

We thank the reviewer for these useful comments. The choice of not using softmax was indeed motivated by ease of implementation in the circuit design. The choice of the nonlinear function used instead was motivated by efficiency in the circuit design: the charge-to-pulse circuit not only has the function of applying a nonlinearity, but it also integrates incoming signals that are temporally encoded, and it transmits the information to the next stage. We added a sentence

in the introduction to clarify this point: “*Additionally, the normalization in softmax requires summing across all input elements, requiring global connections with an increased hardware complexity scaling with the sequence length [45, 46]. In our system, the activation function is instead operated element-wise with charge-to-pulse circuits implementing HardSigmoid functions.*”.

We realized that the name “ReLU” was not entirely appropriate since the function saturates also to the upper bound. Therefore we apologize for the confusion on terminology, and have replaced the name ReLU with Hard Sigmoid in the text.

Here, the choice of element-wise nonlinearity was not designed to decompose the attention computation as in linear attention. Despite the fact that element-wise nonlinearity does not have the normalization factor of softmax, which is useful during training to avoid under or overflow, other nonlinearities such as ReLU² ¹ or sigmoid^{2,3,4} have been shown to be efficient even on large 7 billion parameters models. Moreover, even though softmax is still the most commonly used nonlinearity, it has been shown that the normalization in softmax can bring negative issues such as attention sink which focus attention on non informative tokens², or information loss in sliding window attention³. We added in the introduction the sentence: “*In particular, Sigmoid-based attention has been shown to match Softmax-based attention on models up to 7 billion parameters large [48]. Recent works show that Sigmoid-based attention could even have better information retention than Softmax-based ones [49, 50].*”.

References:

¹ Ma, X. et al. Mega: Moving Average Equipped Gated Attention. Preprint at <https://doi.org/10.48550/arXiv.2209.10655> (2023).

² Gu, X. et al. When Attention Sink Emerges in Language Models: An Empirical View. Preprint at <https://doi.org/10.48550/arXiv.2410.10781> (2025).

³ Fu, Z. et al. Sliding Window Attention Training for Efficient Large Language Models. Preprint at <https://doi.org/10.48550/arXiv.2502.18845> (2025).

⁴ Ramapuram, J. et al. Theory, Analysis, and Best Practices for Sigmoid Self-Attention. Preprint at <https://doi.org/10.48550/arXiv.2409.04431> (2025).

2. Flash Attention is an optimized computational method for the self-attention mechanism on GPU hardware. Its key insight is that, in most cases, the bottleneck of the self-attention mechanism lies in memory limitations. As a memory-compute integrated design for linear attention, I do not see the advantages of this approach. Additionally, the authors should compare the proposed method with Flash Attention in terms of computational latency and energy consumption.

Answer:

Indeed, the bottleneck of the self-attention mechanism lies in memory limitations, but in particular in data transfer between High Bandwidth Memory and SRAM memory. On GPUs,

each inference requires many of these data movements because the capacity of SRAM-based caches is very small due to the large area footprint of SRAM technology. The Gain Cells-based storage allows denser memory integration compared to SRAM, and the memory-compute integrated design allows the system to compute directly without data transfer, except for the write operation which is operated only once per token and per cell, thus reducing the overhead. To clarify the distinction with conventional GPU cache and our implementation, we added the sentence: *“and can be manufactured with very small feature sizes, achieving higher density than SRAM, and also support 3D integration, which can further reduce effective area requirements [40, 41]”* in the introduction.

We would like to clarify that our approach is not an implementation of linear attention. Linear Transformers¹ aim to compress the past states in a fixed size memory. On the other hand, our self-attention is similar to conventional Transformers, as the two dot-products of our system compare explicitly the present query with all keys and values from different time steps. Even without Softmax, the projections are mixed through the dot-products. To discuss the distinction with linear Transformers and Transformers’ alternatives, in the introduction we added the sentence: *“To mitigate this bottleneck, a wide body of literature explores resource-efficient algorithms [9]. Alternative architectures to Transformers with linear time complexity are developed to improve long sequence processing efficiency [10–13]. However, because of their parallelizable training, to this day Transformers are still favored over models such as Mamba [13].”*.

In the comparison between GPU energy consumption and latency and our system (see figure 5 (c and d)), the measurements in GPU were already realized using FlashAttention-2, as we used the function of Pytorch `torch.nn.functional.scaled_dot_product_attention` (see our github repository https://github.com/NathanLeroux-git/GainCellAttention/blob/main/tests_divers/test_speed_energy.py) which utilizes this optimized method (see https://pytorch.org/docs/stable/generated/torch.nn.functional.scaled_dot_product_attention.html#torch-nn-functional-scaled-dot-product-attention). We thank the reviewer for the opportunity to clarify that point. The revised sentence in the method section 4.5 now reads: *“we perform ten runs of 1024 steps of auto-regressive token generation with twelve attention heads using the method FlashAttention-2 [72], which optimizes attention computation in GPUs.”*. Additionally, to permit the simulation and training of larger models, we implemented an optimized Triton kernel for GPU computation that is inspired by FlashAttention. In the method section 4.1, we added the sentences: *“To speed up our training process, we used the library Triton [62] to incorporate our simulations into an adapted version of the Flash Attention algorithm [63], which optimizes GPUs resources. This method led to a factor 5 latency reduction during training.”*.

References:

¹ Katharopoulos, A., Vyas, A., Pappas, N. & Fleuret, F. Transformers are RNNs: Fast Autoregressive Transformers with Linear Attention. Preprint at <http://arxiv.org/abs/2006.16236> (2020).

3. What does the "public software model" in Table 1 refer to? If it refers to GPT-2, the reported performance metrics are inconsistent with the original paper. For instance, the GPT-2 117M

model achieves an accuracy of 45.99 and a perplexity of 35.13 on LAMBADA as stated in the original paper, whereas the authors report results of 32.56 and 40.06, respectively.

Answer:

Yes, "public software model" refers to GPT-2. We thank the reviewer for pointing out the inconsistency in the LAMBADA evaluation. The authors of the original paper used a specific evaluation method which is not public. Based on replication efforts in the community (<https://github.com/openai/gpt-2/issues/131#issuecomment-497136199>, <https://github.com/EleutherAI/lm-evaluation-harness/issues/350>) it seems that the approach in the original paper only considers the first token of the last word and additionally they introduce a list of "stopwords" that are not considered and essentially removed as possible output. We, on the other hand, presented the raw results as obtained from considering the entire last word. In the original paper, without the additional methodology, they report an accuracy of 52.66 % for GPT-2-XL in the text, which matches very closely with the score we obtain with our evaluation strategy, 51.21%. We replicated the custom evaluation strategy of the original paper and now obtain results closely matching the results in the original paper for the public GPT-2 (acc: 45.96 % vs 45.99 %, ppl: 35.15 vs 35.13) and GPT-2-XL (acc: 63.87 % vs 63.24 %, ppl: 9.68 vs 8.63) models. We now use this strategy for all models on the LAMBADA benchmark and have updated the results table accordingly.

4. Using sliding window attention may lead to suboptimal performance. I strongly recommend that the authors discuss the use of current mainstream block-based self-attention methods (such as the chunk-wise approach in Flash Attention) to optimize the proposed method.

Answer:

Our sub-tiling method, shown in Figure 3 (a), is very similar to the chunk-wise approach in Flash Attention: the number of compared past tokens is equal to the product between the number of arrays and the number of columns in each array. The results of the different arrays are accumulated through digital adders. In Flash Attention, the tokens are also divided into different blocks, the computation is done separately for each block, and the results are accumulated at the end.

If Flash Attention utilizes the full context with the appropriate number of blocks, we can also increase the window size by increasing the number of arrays if we need a larger attention span, but with an extra area cost. Nonetheless, sliding window attention is a very common approach in the field of Transformers, in which the attention span is equal to the product between the sliding window size and the number of layers^{1,2,3}. To clarify both points, in section 2.3, we added the sentences: "*In sliding Window Attention, the maximum attention span is equal to $L(M - 1) + 1$ [51]. Therefore, in the presented architecture, the maximum attention span can be increased by increasing the number of sub-tiles. However, this leads to additional area footprint scaling linearly with the sliding window dimension, and additional latency as each digital adder requires one clock cycle.*".

Additionally, to increase computing speed to train our models, which was required to scale to a 1.5 billion model, we developed a Triton⁴ adaptation of the Flash Attention algorithms which simulates our hardware. In the methods 4.1, we added the paragraph: "*To speed up our training process, we used the library Triton [62] to incorporate our simulations into an adapted*

version of the Flash Attention algorithm [63], which optimizes GPU's resources. This method led to a factor 5 latency reduction during training.”.

References:

¹ Jiang, A. Q. et al. Mistral 7B. Preprint at <https://doi.org/10.48550/arXiv.2310.06825> (2023).

² Xiao, G., Tian, Y., Chen, B., Han, S. & Lewis, M. Efficient Streaming Language Models with Attention Sinks. Preprint at <http://arxiv.org/abs/2309.17453> (2023).

³ Fu, Z. et al. Sliding Window Attention Training for Efficient Large Language Models. Preprint at <https://doi.org/10.48550/arXiv.2502.18845> (2025).

⁴ Tillet, P., Kung, H. T. & Cox, D. Triton: an intermediate language and compiler for tiled neural network computations. in Proceedings of the 3rd ACM SIGPLAN International Workshop on Machine Learning and Programming Languages 10–19 (Association for Computing Machinery, New York, NY, USA, 2019). doi:10.1145/3315508.3329973.

5. The authors claim to optimize large language models; however, the experiments are conducted using a GPT-2 model with only 124 million parameters (0.124B). Given the scaling laws and potential outliers, I am concerned that the proposed architecture may suffer significant performance degradation when scaled to models with billions of parameters. I recommend that the authors validate their method on larger models, such as LLaMA2-7B, to better assess its scalability and effectiveness.

Answer:

We thank the reviewer for the remark, and acknowledge the importance of scaling to larger networks, which we have been able to investigate. Due to academic computing resource limitations, we were not able to train up to 7B models, which would have increased the computation requirements 70 times compared to the previous effort. Nonetheless, we have been able to train GPT-2-XL which has 1.5 billion parameters, and allows for scaling analysis. In the section results 2.5, we added the paragraph: *“To evaluate scalability on larger models, we applied the same training methodology to GPT-2-XL, which contains 1.5 billion parameters. While its performance is slightly lower than the software-only baseline, it significantly outperforms the smaller GPT-2 model. Moreover, the gap with its software counterparts is not larger than for GPT-2. This demonstrates that increasing model size with our hardware-based attention mechanism leads to improved accuracy. We attribute the remaining performance gap to quantization effects and the use of HardSigmoid activation, which clamps values and therefore can cause vanishing gradients when inputs fall outside the activation range.”* and updated the corresponding benchmark table.

Reviewer #2 (Remarks on code availability):

Codes are nice for readers to reproduce, and the detailed readme file is provided.

Answer: We thank the reviewer for this comment.

Reviewer #2 (Remarks on figshare data availability):

These data are in agreement with the claims of the Article.

Answer: We thank the reviewer for this comment.

Reviewer #3 (Remarks to the Author):

This paper proposes a custom self-attention in-memory computing (IMC) architecture to address the energy and latency bottlenecks in large language models (LLMs). Specifically, the authors leverage capacitor-based analog gain cells as memory and computation units to store key-value (KV) projections and perform parallel dot-product computations central to the self-attention mechanism. A hardware-aware adaptation algorithm enables pre-trained language models (e.g., GPT-2) to operate effectively on this non-ideal analog hardware, without requiring retraining from scratch. However, the architecture's reliance on analog components introduces trade-offs in accuracy and generalizability, while the justified approach on the attention mechanism limits its broader applicability. The reliance on simulations rather than physical implementations further diminishes the immediate impact of the work. As a result, I cannot recommend the manuscript for publication, at least not in the present form.

1) It should be noted that the GPT-2 model selected by the authors is described as having 117 million parameters in the original paper [1], which is inconsistent with the description provided by the authors in section 4.2 of this paper. Additionally, in the original paper, the 12-layer GPT-2 has a perplexity (PPL) of 35.13 and an accuracy (ACC) of 45.99 on LAMBADA, while the perplexity on WikiText2 is 29.41. These numbers differ significantly from the data presented by the authors in TABLE 1. The authors have seemingly diminished the performance of the baseline considerably. Compared to the original performance, there is a considerable gap in task performance with the proposed method in this paper.

Reference:

[1]Radford, Alec, et al. "Language models are unsupervised multitask learners." OpenAI blog 1.8 (2019): 9.

Answer:

We thank the reviewer for the different useful comments. To demonstrate further generalizability and scalability, we have trained a GPT-2-XL model with 1.5 billion parameters (~10 times larger than previously), and updated the results section as well as the table.

We also thank the reviewer for pointing out the inconsistency on model parameters. However, the 117 million parameters in the original paper is most likely an error, as the GPT-2 model release post written by OpenAI (the authors) states that the model has 124 million parameters (see <https://huggingface.co/openai-community/gpt2>). We also made thorough verification on the model from the OpenAI GPT-2 checkpoint, and it was indeed 124 million parameters.

We apologize for the inconsistency on evaluation results. The authors of the original paper used a specific evaluation method which is not public. Based on replication efforts in the community (<https://github.com/openai/gpt-2/issues/131#issuecomment-497136199>,

<https://github.com/EleutherAI/lm-evaluation-harness/issues/350>) it seems that the approach in the original paper only considers the first token of the last word and additionally they introduce a list of “stopwords” that are not considered and essentially removed as possible output. We, on the other hand, presented the raw results as obtained from considering the entire last word. In the original paper, without the additional methodology, they report an accuracy of 52.66 % for GPT-2-XL in the text, which matches very closely with the score we obtain with our evaluation strategy, 51.21 %. We replicated the custom evaluation strategy of the original paper and now obtain results closely matching the results in the original paper for the public GPT-2 (acc: 45.96 % vs 45.99%, ppl: 35.15 vs 35.13) and GPT-2-XL (acc: 63.87 % vs 63.24 %, ppl: 9.68 vs 8.63) models. We now use this strategy for all models on the LAMBADA benchmark and have updated the results table accordingly. For WikiText2 the exact evaluation method used in the original GPT-2 paper is unclear (<https://github.com/openai/gpt-2/issues/78>, <https://github.com/karpathy/llm.c/pull/276>). Therefore, we report perplexity scores normalized by word count of the original text as reported by the lm-evaluation-harness (<https://github.com/EleutherAI/lm-evaluation-harness>), which is widely used in the community. We added a sentence in the methods 4.3 to clarify the evaluation choice: “*For Wikitext-2 we report perplexity scores normalized by the word count in the original text.*”

2) Due to hardware constraints, the authors have made some adjustments to the transformer algorithm, replacing the softmax function after $Q \cdot K^T$ with ReLU. In the original transformer, the softmax function not only normalizes the attention between tokens but also introduces competition between the attentions of different tokens. By replacing it with ReLU, the authors have only removed negative attention values, which leads to two results: i) The possibility of extremely large values; ii) Each attention value becomes an independent attention. The authors conducted experiments only on the 12-layer GPT-2, and according to the original data of GPT-2, there is still a performance gap with the proposed method. This gap is expected to widen when dealing with larger models.

Answer:

We acknowledge that softmax is still the most commonly used nonlinearity, and the normalization is useful for training stability. However, it has been shown that the normalization in softmax can also bring negative issues such as attention sink which focus attention on non informative tokens¹, or information loss in sliding window attention². In other words, the competition between tokens is not always the most desirable behavior, as it can hinder useful information. In the introduction, we added the sentence: “*In particular, Sigmoid-based attention has been shown to match Softmax-based attention on models up to 7 billion parameters large [48]. Recent works show that Sigmoid-based attention could even have better information retention than Softmax-based ones [49, 50].*”

We have realized calling the nonlinear activation in our system “ReLU” was not wholly accurate, since the function in practice saturates to an upper bound (see figure 1 (g)). Physically, the limit of the nonlinear function corresponds to the maximum pulse length emitted by the charge-to-pulse circuit. Thus, there is no possibility for extremely large values. We have replaced the name ReLU with the more suited name HardSigmoid in the text.

The attention values are not independent since they are mixed by the two dot-products. Each inference step has attention scores for the different previous tokens using stored keys K . The second dot-products then multiplies the attention scores with stored values V of the different previous tokens.

To demonstrate scalability, we trained GPT-2-XL which has 1.5 billion parameters. In the section results 2.5, we added the paragraph: “*To evaluate scalability on larger models, we applied the same training methodology to GPT-2-XL, which contains 1.5 billion parameters. While its performance is slightly lower than the software-only baseline, it significantly outperforms the smaller GPT-2 model. Moreover, the gap with its software counterparts is not larger than for GPT-2. This demonstrates that increasing model size with our hardware-based attention mechanism leads to improved accuracy. We attribute the remaining performance gap to quantization effects and the use of HardSigmoid activation, which clamps values and therefore can cause vanishing gradients when inputs fall outside the activation range.*” and updated the corresponding benchmark table.

References:

¹ Gu, X. et al. When Attention Sink Emerges in Language Models: An Empirical View. Preprint at <https://doi.org/10.48550/arXiv.2410.10781> (2025).

² Fu, Z. et al. Sliding Window Attention Training for Efficient Large Language Models. Preprint at <https://doi.org/10.48550/arXiv.2502.18845> (2025).

3) The authors propose using sliding window attention to replace causal attention, which also achieves the masking functionality in decoder-only LLMs. However, it also results in the loss of a significant amount of needed attention, such as the attention between Q3 and K0 in fig 2.a. The impact of such lost attention will accumulate with larger models, potentially having a huge effect on task performance when dealing with larger models.

Answer:

Sliding window attention indeed does not utilize the full context, but it is still a very common approach in the field of Transformers, in which the attention span is equal to the product between the sliding window size and the number of layers ^{1,2,3}. The impact of sliding window rather has a tendency to reduce for deeper models, as stacking layers allows the aggregated representations from neighboring tokens to increase the receptive field.

We can also increase the window size by increasing the number of arrays if we need a larger attention span, but with an extra area cost. To clarify this point, we added the sentence “*In sliding Window Attention, the maximum attention span is equal to $L(M - 1) + 1$ [51]. Therefore, in the presented architecture, the maximum attention span can be increased by increasing the number of sub-tiles. However, this leads to additional area footprint scaling linearly with the sliding window dimension, and additional latency as each digital adder requires one clock cycle.*” in section 2.3.

References:

¹ Jiang, A. Q. et al. Mistral 7B. Preprint at <https://doi.org/10.48550/arXiv.2310.06825> (2023).

² Xiao, G., Tian, Y., Chen, B., Han, S. & Lewis, M. Efficient Streaming Language Models with Attention Sinks. Preprint at <http://arxiv.org/abs/2309.17453> (2023).

³ Fu, Z. et al. Sliding Window Attention Training for Efficient Large Language Models. Preprint at <https://doi.org/10.48550/arXiv.2502.18845> (2025).

4) The authors mention that resistive losses in interconnects, known as IR drop issues, can cause reduced accuracy in large-scale analog crossbar arrays. They mitigate this by limiting the size of their gain cell arrays. However, how does this size limitation impact the scalability of the proposed architecture? This could potentially limit the system's ability to handle larger models or more complex tasks, which could affect its broad applicability and usefulness in real-world applications. And it could also introduce additional complexity and potential overheads, affecting the overall efficiency of the architecture.

Answer:

To face the issue of limited array size, we use a sub-tiling strategy that allows us to stack multiple arrays, as shown in figure 3 (a). This method can either be used to increase the sliding window size, or to increase the embedding size (although embedding sizes larger than 64 or 128 per head are rarely used in Transformers). At this stage, we don't see limiting factors other than area footprint, concerning complexity in increasing the number of stacked arrays. However, increasing the dimensions introduce overheads, as the results of the different arrays must be summed by digital adders. Additional IMC sub-tiles will not add extra latency since they operate in parallel, but the sequential digital addition operation at the final step will increase latency by one clock cycle for any additional sub-tile. For instance if we want to increase the sliding window dimension from 1024 to 2048 we would need 16 additional sub-tiles, therefore 16 additional ns, hence in total 81 ns instead of 65 ns. To clarify this point, we added the sentence "*in the presented architecture, the maximum attention span can be increased by increasing the number of sub-tiles. However, this leads to additional area footprint scaling linearly with the sliding window dimension, and additional latency as each digital adder requires one clock cycle.*" in section 2.3.

5) The authors' architecture utilizes storage capacitors in its design, which are known to have retention issues leading to stored value decay over time. How does this leakage impact the long-term performance and reliability of the proposed system? While this is a less critical limitation in short-sequence processing, if not appropriately managed, limited retention could lead to a gradual degradation in system performance, potentially affecting the accuracy of computations and limiting the system's effectiveness in long-term sequence processing or scenarios requiring extended operations.

Answer:

We thank the reviewer for the opportunity to further study this aspect. We answer through a theoretical and an experimental perspective, and have added this to the manuscript.

In particular, we have added a mathematical analysis in the supplementary material which shows the similarity of the decay induced by the capacitor's leakage with the positional embedding method called ALiBi¹. ALiBi is a method used to introduce relative positional information in long sequences. We compare the decay amplitude in our system with the one in ALiBi, and show that ALiBi decay is even larger, while ALiBi allows generalization to larger sequences. This study shows that even larger decays than the one in our system can have a positive effect.

However, the decay factor, which is the ratio between latency and retention time, depends on both the type of gain cell and the latency of the full architecture, which is hard to determine at this stage. Therefore, to study the effect of decay in multiple configurations, we varied the value of the decay factor for our trained hardware model. The results are shown in the new figure 6 in the supplementary material, and commented in the new section 4 of the supplementary material. Results show that the accuracy starts decreasing when the decay factor is larger than 10^{-3} . This means that if the retention time of the Gain Cells is 10 s, the full architecture latency should not exceed 10 ms (in comparison, the latency per token reported on a GPU H200 on LLAMA 2 70B is 0.1 ms^2). However, training with larger decay could also mitigate negative effects, potentially helping to generalize to larger decays.

References:

¹ Press, O., Smith, N. A. & Lewis, M. Train Short, Test Long: Attention with Linear Biases Enables Input Length Extrapolation. Preprint at <https://doi.org/10.48550/arXiv.2108.12409> (2022).

² NVIDIA Blackwell Platform Sets New LLM Inference Records in MLPerf Inference v4.1. NVIDIA Technical Blog <https://developer.nvidia.com/blog/nvidia-blackwell-platform-sets-new-llm-inference-records-in-mlperf-inference-v4-1/> (2024).

Reviewers' Comments:

Reviewer #1:

Remarks to the Author:

I want to thank the authors for addressing some of my concerns. The manuscript was improved, but some major issues remain:

The new results on GPT-2-XL are not very convincing and the interpretation flawed. For example, the accuracy on ARC-E dataset drops from 58.29 to 53.79 with the proposed HW model, while the accuracies were quite similar for the GPT-2 model. Hence, there is a significant drop of 4.5, but still the authors conclude "... the gap with its software counterparts is not larger than for GPT-2". However, I am not convinced that the proposed simplified replacement of the attention mechanisms can be scaled to state-of-the-art model sizes (other than the authors claim "This demonstrates that increasing model size with our hardware-based attention mechanism leads to improved accuracy"). The manuscript does not provide a detailed mathematical analysis of the impact of the proposed method. In that same paragraph on page 11, the authors claim "We attribute the remaining performance gap to quantization effects and the use of HardSigmoid activation, which clamp values and therefore can cause vanishing gradients when inputs fall outside the activation range", without a justification. It would be necessary to conduct additional experiments, and rigorous theoretical modelling of the proposed approximations to quantify the impact. In addition, I would recommend testing the model on at least one additional model size of > 3 B parameters. Such an experiment would more clearly show the trend and justify the claims made throughout the paper. Without such a justification, the many claims about the scalability of their method need to be dropped.

In their added text on page 2 the authors claim: "However, because of their parallelizable training, to this day Transformers are still favoured over models such as Mamba [13]", but that is wrong. Mamba is highly optimized for efficient use of hardware and has demonstrates excellent scaling properties. In fact, the parallel scan algorithm that can be efficiently executed in parallel on GPU hardware was one of the main innovations underlying MAMBA (also other SSM architectures share this property). The reviewer acknowledges that transformers are still heavily used despite their unfavourable N^2 time complexity, but parallelizable training is not the issue here. Please rephrase this paragraph to resolve this very misleading statement.

The system level implementation outlined in 5.1 is somewhat contrived and incomplete. The proposed method uses an external hardware block that implements the required intermediate MLP layers, while only self-attention is done on the chip. The main advantage of 3D stacking is the reduction of wiring lengths. However, this advantage is completely lost here, since attention layers are simply stacked on top of each other, while the required intermediate MLP parts are left externally. Also, the

authors show in their estimation of the energy budget in section 5.2 that only a single of these layers can be activated at a time. This seems quite wasteful, in light of the 3D stacking. If the energy budget is prohibitive of parallel pipelining (e.g. batch-parallel execution of the multiple prompts), this seems to suggest that the architecture is ill-suited for 3D stacking altogether.

Remarks on code availability:

The added comments are helpful, thanks.

Answer:

Model scalability:

We thank the reviewer for the thoughtful feedback. To clarify the apparent drop on the ARC-E dataset for GPT-2-XL, we added two columns to the table for average accuracy and average perplexity, and included in parentheses the differences between the small and large models. These aggregated results reveal that the mean performance gain from GPT-2 to GPT-2-XL with our hardware attention is comparable to that of the software baseline. Furthermore, we clarify that the public GPT-2-XL checkpoint benefits from an undisclosed pre-training budget, whereas our hardware model was trained from scratch for only 13 000 iterations under a limited compute budget. For a fairer comparison, we also included results for an equivalent GPT-2-XL software model trained from scratch (using softmax, without quantization). These matched-iteration results show that our hardware model does not underperform the software baseline. Together, these findings indicate that performance differences arise from training budget rather than from the HardSigmoid activation or quantization. We have updated the Results section accordingly.

The reviewer suggests evaluating our approach on a > 3 B-parameter model; however, training such a model to convergence would require on the order of 100,000 GPU-hours, incurring substantial financial and time costs, which explains why most academic works experiment with models of at most 1 B parameters.

Finally, regarding the request for additional mathematical background on non-softmax attention, we note that this topic has been thoroughly studied in the literature, particularly in Ramapuram et al. (2024), which provides rigorous proofs of the universality of piecewise-sigmoid attention and demonstrates that, despite some training instability at large scales, a 7 B-parameter model with regularization such as QK-normalization can match softmax-based performance.

Comparison with Mamba:

We acknowledge the reviewer's valid point regarding the efficiency of Mamba and

other SSM architectures, particularly the use of parallel scan algorithms that significantly improve hardware utilization on GPUs. While the parallel scan technique relies on a tree-like structure requiring $O(\log(N))$ sequential steps—thus not being fully parallel in the strictest sense—we recognize that this distinction may be overly technical and that our original phrasing could indeed be misleading. We have therefore rephrased the sentence in question to avoid overemphasizing the lack of parallelism and instead highlight other important limitations of SSM-based models, such as their less favorable scaling with model size in terms of both computational efficiency and accuracy. Furthermore, many of the largest and most successful SSM models to date, such as Samba (L. Ren et al 2024. [arXiv.org https://arxiv.org/abs/2406.07522v3](https://arxiv.org/abs/2406.07522v3)), are hybrid architectures that interleave state-space layers with attention mechanisms. We rephrase the sentence the reviewer was pointing to by: “However, Transformers continue to exhibit more stable training at scale than alternatives such as Mamba \cite{gu_mamba_nodate}, which contributes to their ongoing dominance despite the efficiency of state-space models.”.

System level 3D integration:

We believe that our new section 5.2 in the supplementary material might have led to some confusion.

The reviewer points out as an issue that our MLP layers are left externally and not integrated in 3D. We would like to clarify that there is no obstacle to integrate MLP layers implemented by memristors in 3D in our architecture. We simply did not show that in our sketch because we thought it might be confusing for the reader since this article only focuses on attention. To clarify this point, we added the sentence: “In this study, we focus exclusively on the computation of the attention mechanism. However, the proposed architecture is also compatible with \ac{IMC} arrays capable of performing the linear layers required for a complete network. These linear layers could be integrated in a 3D-stacked configuration using non-volatile memory technologies, as demonstrated in \cite{buchel_efficient_2025}”.

The reviewer seems to doubt the suitability of 3D stacking in terms of speed or energy consumption with the arguments that only one layer is activated at the time. This implementation does not impact speed, and we did not make any claim about 3D integration increasing parallelism. We simply claim that it reduces area footprint. Our system design activates only one layer at the time for better thermal management, which was the concern that led to integrate this new section.

The reviewer argue that our system would be ill fitted for 3D stacking on the basis that it would not support batch inference. However, using 3D stack for batch inference would require redundant memories over the different layers. On the contrary, our approach does not require redundant memory, and even allows us to reuse some of the CMOS hardware. In essence, our 3D system design reduces the area footprint by leaving latency unchanged compared to the 2D implementation.

To clarify this point, we added the sentence: “In this section, we propose a hardware design that efficiently implements multiple Transformer layers and attention heads using 3D-stacked OSFET-based \ac{IMC} arrays, where stacking minimizes area footprint and sequential physical layer activation helps manage thermal constraints.”.

Reviewer #2:

Remarks to the Author:

Thank you for addressing my previous comments; they have resolved most of my concerns. I still have reservations about Comment 2—the details of the latency and energy-consumption comparisons on GPUs. I believe the authors should provide (1) a broader range of GPU models, (2) a larger context window (e.g., 10–2048 tokens), and (3) a wider set of model baselines. Because latency and energy measurements are independent of model accuracy, the authors could also evaluate larger LLaMA variants. Strengthening these comparisons would better highlight the contribution of the work.

The experimental evaluation should be performed on mainstream NVIDIA GPUs from the Ampere and Hopper generations that provide native support for Flash-Attention and Flash-Attention V2 (e.g., the A100, H100, and H200). We are, moreover, keen to observe the method’s behaviour on alternative accelerators, such as Huawei Ascend 910C and AMD Instinct MI300X. The benchmark models ought to span a wide parameter spectrum approximately 1 B to 70 B parameters. Suitable model families include Qwen, LLaMA, and DeepSeek, and sequence lengths should range from 128 to 1 024 tokens. We strongly encourage the authors to conduct the above experiments to furnish a more comprehensive comparison of their approach against prevailing parallel-computing architectures.

Remarks on code availability:

The code provides a README file with enough instructions, and readers can easily run it. I checked the code but didn't run it.

Remarks on figshare data availability:

They are consistent with the results in the article.

Answer:

We thank the reviewer for the useful comments and suggestions.

To improve the comparison with other technologies, experimenting on other hardware is indeed valuable. However, we can only experiment on the hardware we have at our disposal, or otherwise it would drastically increase delays. We were

therefore able to reproduce the energy and latency measurement on a Nvidia H100, as the reviewer suggested, which is a valuable experiment as H100 is a very well-known and used GPU architecture for neural network training and inference. We updated the figure 5, the Results section, and the Methods section accordingly.

The reviewer suggests to experiment on different window size. The issue is that such experiment would not be very informative, in perspective to the very large required experimental effort and time for the electrical simulations. The energy and latency of our hardware attention simply scales proportionally to the window size, since the window size is determined by the number of sub-tiles. The energy and latency for the 1024 window size are already extrapolated from 64x64 arrays simulations. To clarify this point, in the Methods "Hardware SPICE simulations" we added the sentence: "To extrapolate the energy and latency for a full attention head with a window size of 1024, we multiply the per-sub-tile measurements by 16, reflecting the total number of sub-tiles comprising one attention head in our architecture."

The reviewer suggests to experiment with a wide variety of model parameter range and families. Unfortunately, there is a slight misalignment between this request and our experiments. Because our article only focuses on the scaled dot-product of the attention, our energy consumptions comparison also focuses on this specific module, and we simulate only a single multi-heads attention block, and not the full model. To clarify this point, in the Methods "GPU Attention Latency and Energy Consumption Measurements" we added the sentence "The energy and latency consumption measurement solely focus on attention computation, and for a fair comparison, the linear projections are not implemented in this experiment since they are also not implemented by our hardware architecture ". Therefore, experimenting on models ranging from 1B to 70 B parameters would not lead such different results. Besides, it would also require very expensive resources. Experimenting on a vast variety of model families is also an expensive request, and not really suited since we don't simulate the full models.

Reviewer #3:

Remarks to the Author:

I appreciate the detailed response from the authors and the substantial work to improve the quality of this work. I'm pleased to see that my previous concerns have been addressed in this revision. The authors have carefully revised the manuscript accordingly. I would like to recommend this paper to be accepted for publication.

Answer:

We thank the reviewer for these comments, and are very pleased to see that our efforts were appreciated.